# Language Models can Self-Improve at State-Value Estimation for Better Search

**Ethan Mendes, Alan Ritter**
Georgia Institute of Technology
emendes3@gatech.edu, alan.ritter@cc.gatech.edu

## Abstract

Collecting ground-truth rewards or human demonstrations for multi-step reasoning tasks is often prohibitively expensive, particularly in interactive domains such as web tasks. We introduce Self-Taught Lookahead (STL), a reward-free framework that improves language model–based value functions by reasoning explicitly about state transitions. STL can be viewed as a chain-of-thought analogue of the value iteration algorithm: instead of regressing directly on numeric values, a value LLM is trained to simulate a step of lookahead in natural language—predicting the next action, resulting state, and rationale for its value, thereby refining value estimates without any labeled data. This self-supervised procedure yields more accurate state-value predictions, which in turn enable lightweight search algorithms to expand fewer states while maintaining strong performance. Empirically, STL-trained value models built on moderately sized (8B parameter) open-weight LLMs boost web agent success rates by over 39%, achieving comparable performance with proprietary models. STL also generalizes to multi-hop QA and math puzzles. We find that STL enables small open-source models to guide efficient search, reducing inference costs by integrating explicit reasoning with value learning.

## 1  Introduction

While large language models (LLMs) demonstrate strong reasoning capabilities by generating extended token sequences before answering [56, 22, 40], guiding inference with explicit tree search has the potential to further improve performance on tasks with a structured state space [64, 60]. In this setting, a policy LLM proposes candidate actions, and a value LLM evaluates resulting states to steer the search toward promising trajectories. Figure 1 summarizes the assumptions different LLM-driven search methods make about information available during training and inference. **Reward and Demo Learning** methods [63, 70, 27, 52] assume access to ground-truth reward signals or human demonstrations and optimize the model with reinforcement learning (RL) or imitation learning (IL). In contrast, **Reward-Guided Inference** strategies [74, 49] forego explicit reward during training; they only consult a reward signal at inference time, using it to guide procedures such as LLM-based Monte-Carlo Tree Search (MCTS). However, collecting ground-truth rewards or human demonstrations may not be possible in every environment, and can oftentimes be costly. For example, for web agent tasks, even small-scale data collection can cost thousands of dollars [63]. **Reward and Demo Free** methods [65, 64] relax this assumption as they can operate without access to reward. However, these methods often rely on prompting an off-the-shelf LLM to serve as both the policy and value models during the search process, which constrains performance compared to models specifically tuned for agentic tasks [55].

In this paper, we introduce Self-Taught Lookahead (STL), a **Reward and Demo Free**, self-supervised framework for interactive, multi-step reasoning tasks. Building on evidence that search quality is strongly influenced by the accuracy of the state-value estimator [10, 37], STL improves an

39th Conference on Neural Information Processing Systems (NeurIPS 2025).

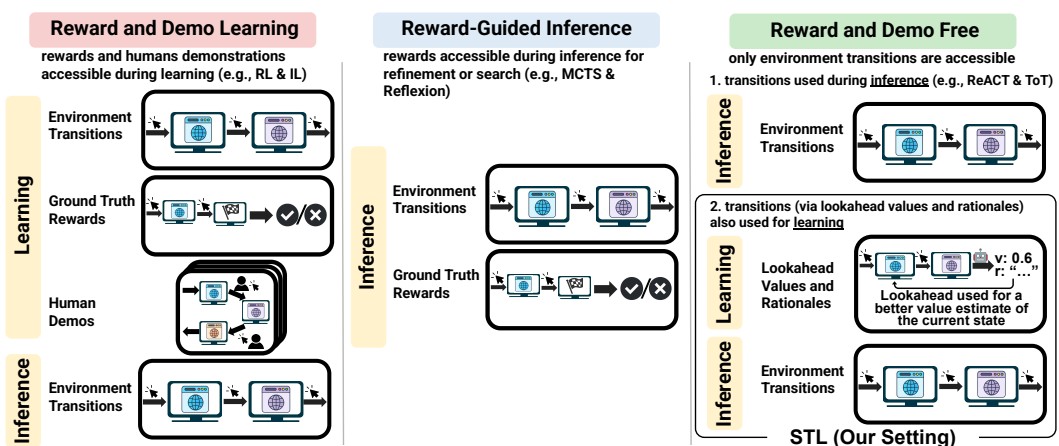

Figure 1: The information accessible during learning and inference across common search settings, exemplified using web tasks. Our Self-Taught Lookahead method is **Reward and Demo Free**, yet is able to self-improve by learning from state transitions in the form of lookahead values and rationales.

LLM-based value function for better search performance. Unlike neural models traditionally used for state-value estimation in the learning to search literature [50], an LLM can leverage both conventional numerical values and natural language reasoning to estimate state values. STL exploits this feature by having an LLM value model learn to better assign values to states based on their expected future utility by constructing and learning from rationales that explicitly capture *state transition dynamics*. For instance, without understanding these transitions, it might not be clear whether a CLOSE (X) button on a website interface exits the current view or the entire workflow in web tasks [23]. Learning better state-value estimates from state transition dynamics is especially well-suited for self-improvement in agentic tasks, where the environment directly provides transition outcomes. As a result, our approach requires neither ground truth rewards nor human demonstrations.

STL (Figure 2) begins by generating self-improvement data through a single step of lookahead within a tree search. Analogous to the Bellman update, this lookahead refines the estimated value of a state by leveraging information about potential future states. However, unlike classical reinforcement learning (RL) methods, which rely on explicit environment rewards, STL uses a large language model (LLM) to estimate state values. Specifically, during STL, a value LLM is fine-tuned to reason about the utility of a state by predicting the next best action, its resulting state, and a corresponding rationale for the value of that state. During training, the model is fine-tuned using rollouts of states and actions within the environment. At inference time, instead of taking a step of lookahead in the environment, the improved value model *simulates* a step of lookahead to provide more accurate value judgments.

By representing the lookahead process in natural language rather than regressing solely on value estimates, STL takes advantage of LLMs' strong generalization to unseen tasks via learned textual reasoning [42, 68]. For instance, our results (§4) on web agent tasks from WebShop [63] demonstrate that tree search with an STL fine-tuned `llama-3.1-8b-instruct` value model improves performance by $39\%$ or more compared to the base `llama` model. Furthermore, STL matches the performance of search with a base `gpt-4o` value model and even achieves comparable results to **Reward-Guided Inference** methods such as LATS [74] on unseen tasks. We show that these results also hold for math puzzles and multi-hop question answering [64, 62]. Finally, by enabling the use of a small open-source value model during inference, STL leads to significant cost reduction when agents are deployed. Through an efficiency analysis (§5), we find that STL costs $5\times$ less than a similarly performing `gpt-4o` value model. As STL produces more accurate state-value estimates, the resulting value models can effectively guide search algorithms that expand fewer states, while maintaining strong performance.

## 2   Background: Guiding Tree Search with Language Models

Within a state space $\mathcal{S}$, the goal of the tree search is to reach a desired state $s_* \in \mathcal{S}$ from an initial state $s_0 \in S$, where $s_*$ is determined based on the natural language task $x$, with $x \in \mathcal{X}$, the set of

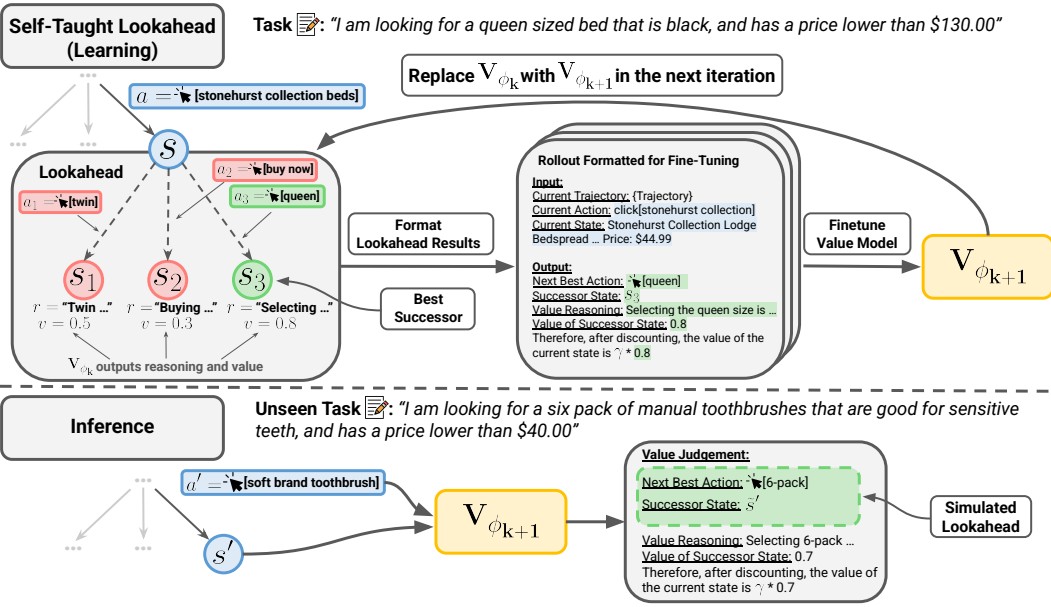

Figure 2: Self-taught lookahead self-improves the value model by learning from state-transition dynamics. During the data generation phase (**top left**), tree search is used to discover diverse states. For every observed state $s$ encountered during the search, successor states are expanded using base policy $\pi_\theta$ and the current value model $V_{\phi_k}$, and a textual training example is formed using verbal representations of the next best action and successor state, as well as $V_{\phi_k}$'s outputted value reasoning ($r$) and numerical value ($v$) discounted by $\gamma$ (**top middle**). These examples are used to fine-tune $V_{\phi_{k+1}}$, which will be used in the next iteration of the algorithm (**top right**). Value models learned during STL can be used to evaluate unseen states encountered during search on unseen tasks by simulating a step of lookahead, including the next best action and the best successor state $\tilde{s}'$ (**bottom**).

all possible tasks. A state $s_i \in S$ might be a step in a reasoning chain or an intermediate webpage in a web navigation task. While the algorithmic details vary based on the tree search method e.g., breadth-first search (BFS) or MCTS, to adapt these methods to utilize language models (LMs), we simply need to define how new states (*successors*) are generated and evaluated.

**Action generation.**    Given a trajectory of $i + 1$ states during the search process, candidate actions $a_i^{(j)}$ in the action space $\mathcal{A}$ are sampled using an LLM-based policy $\pi_\theta$:

$$a_i^{(j)} \sim \pi_\theta(a_i|x, s_0, \ldots, s_i), \forall j \in \{1, \ldots, B\} \tag{1}$$

where $B$ is the branching factor or the number of specified candidate actions. These sampled actions constitute the set $A_{s_i}$. We denote the transition function as $T : \mathcal{S} \times \mathcal{A} \to \mathcal{S}$, so for an action $a_i$, $s_{i+1} = T(s_i, a_i)$.

**State evaluation.**    A *value* $v_{s_i|x}$ and a *rationale* for the value $r_{s_i|x}$ of a state $s_i$ is generated using a LLM-based value model $V_\phi : \mathcal{X} \times \mathcal{S}^* \to \mathbb{R} \times \mathcal{L}$, where $\mathcal{S}^*$ is the set of all finite sequences over the state space $S$ and $\mathcal{L}$ is the space of natural language sequences:

$$(r_{s_i|x}, v_{s_i|x}) \sim V_\phi(x, s_0, \ldots, s_i) \tag{2}$$

Note that because $V_\phi$ is an LLM, it is usually prompted in a chain-of-thought [56] manner to first generate $r_{s_i|x}$ and then conditionally generate $v_{s_i|x}$ based on the rationale. For notational simplicity, will denote these two generated entities as $r_{s_i|x} \sim \text{Rat}(V_\phi(x, s_0, \ldots, s_i))$ and $v_{s_i|x} \sim \text{Val}(V_\phi(x, s_0, \ldots, s_i))$. While previous work [74, 32, 64, 66] do generate rationales during state evaluation, usually they are generated purely to leverage the performance improvements of LLMs when asked to rationalize and are subsequently discarded. As described in §3, inspired by self-taught reasoning [68], we explicitly use these rationales during self-improvement by fine-tuning on them along with the value to learn how to better estimate state values in a given domain.

# 3  Better State-Value Estimation with Self-Taught Lookahead

In this section, we present our proposed self-taught lookahead method. §3.1 and §3.2 describe the STL improvement procedure, while §3.3 explains how a STL-improved value model operates during inference. See Figure 2 and Algorithm 1 in Appendix C for an overview of the method.

## 3.1  Generating Rollouts

STL assumes a static policy model $\pi_\theta$ and only trains the value model through one or more iterations of self-improvement. We denote the value model initialized with a base LLM $V_{\phi_0}$ and the value model used in a subsequent iteration $k$ to assign values and generate rationales $V_{\phi_k}$.

An iteration $k$ of STL starts with a dataset $\mathcal{D}_{\texttt{rollout}_k} \subset \mathcal{X}$ of natural language tasks for the current iteration. For each $x_i \in \mathcal{D}_{\texttt{rollout}_k}$, we roll out the search tree using $\pi_\theta$ and $V_{\phi_k}$. Using tree search enables us to collect a diverse set of states so that the value model trained on these states' values can better generalize to unseen states and tasks. We demonstrate this generalization in §4. When visiting state $s_j$ on the trajectory $\{s_0, \ldots, s_j\}$ during tree search, we compute $s_j$'s **lookahead value**, $y_{s_j}$:

$$y_{s_j} \leftarrow \gamma \max_{a \in A_{s_j}} \left\{ \text{Val}(V_{\phi_k}(x_i, s_0, \ldots, s_j, T(s_j, a))) \right\} \tag{3}$$

where $\gamma$ is the discount factor. Since tasks are episodic, we set $\gamma = 1$ [4, 54, 35]. These lookahead values capture a better estimate of the true value of $s_j$ as they account for $s_j$'s successor states. In §7, we describe how generating and learning from these $y$'s is similar to fitted value iteration.

**Action-outcome rationales.**  However, alone, these lookahead values fail to reflect *why* a given state is valuable as they do not capture **(1)** which action yielded the best (highest value) successor state and **(2)** why the best successor state was assigned a high value by $V_\phi$.

To better capture the state transition dynamics, we also generate **action-outcome rationales** when visiting a state $s_j$. These rationales are of the form "{action} {outcome_state} {value_rationale}" where `action` is selected by the max operator in Equation 3, `outcome_state` is the state observed after taking the `action` in the environment and `value_rationale` is the rationale for the evaluation of this successor state generated by $V_{\phi_k}$. Fine-tuning on these rationales will enable a value model to predict the result of taking an action and incorporate this prediction (lookahead) into the current state's value estimate. Formally, we can define these action-outcome rationales $o_{s_j}$:

$$o_{s_j} \leftarrow a_j^* || s_{j+1}^* || \text{Rat}(V_{\phi_k}(x_i, s_0, \ldots, s_j, s_{j+1}^*)) \tag{4}$$

where $\cdot || \cdot$ denotes concatenation, $s_{j+1}^* \leftarrow T(s_j, a_j^*)$, and

$$a_j^* \in \arg\max_{a \in A_{s_j}} \left\{ \text{Val}(V_{\phi_k}(x_i, s_0, \ldots, s_j, T(s_j, a))) \right\} \tag{5}$$

The training data set for iteration $k$ is thus a set of tuples: $\mathcal{D}_k = (s_k, o_{s_k}, y_{s_k})$. Depending on the task, it might be necessary to automatically filter out tuples that have malformed rationales or account for the same state seen multiple times in different iterations (see Appendices D, E, and F).

## 3.2  Fine-Tuning the Value Model

We start training the new value model $V_{\phi_{k+1}}$ from the initial or base LLM value model $V_{\phi_0}$. We can then train using standard fine-tuning negative log-likelihood loss for the generation of both the action-outcome rationale and the lookahead value ($o_s || y_s$) of the state. We train the value model to generate the rationale before estimating the value. Automatically constructed text or formatting, as seen in Figure 2, is applied for easier learning.

## 3.3  Search after Self-Taught Lookahead

A value model resulting from iteration $k$ of STL ($V_{\phi_{k+1}}$) can directly replace a value model in any search algorithm, such as Greedy Search (§4.1 and §4.2) and BFS (§4.3). As shown in Figure 2 , $V_{\phi_{k+1}}$ *simulates a step of lookahead* for the state $s_n$ i.e. for $(r_{s_n|x}, v_{s_n|x}) \sim V_{\phi_{k+1}}(x, s_0, \ldots, s_n)$,

$$r_{s_n|x} = \tilde{a}_{n+1} || \tilde{s}_{n+1} || \tilde{r}_{s_{n+1}|x} \tag{6}$$

where $\tilde{a}_{n+1} || \tilde{s}_{n+1}$ is a simulated lookahead step and $\tilde{r}_{s_{n+1}|x}$ is its value rationale.

Table 1: Score and success rate (SR) on WebShop. Results marked with † are taken from previous work [74, 47]. Value functions marked with ‡ are fine-tuned. We observe a near 40% improvement in success rate when using the STL value function compared to the `llama-3.1-8b-instruct` base value model in the greedy setting. We compute statistical significance of Reward and Demo Free methods against the underlined results ($^*p < 0.05$, $^{**}p < 0.01$, $^{***}p < 0.001$) using the paired bootstrap test [3]. Best results in the Reward and Demo Free setting are **bolded**.

| Setting | Method | Policy | Value | Mini Test Set (50) | | Full Test Set (500) | |
|---|---|---|---|---|---|---|---|
| | | | | Score ↑ | SR ↑ | Score ↑ | SR ↑ |
| **Reward and Demo Learning** | IL | BERT$^‡$ + BART$^‡$ | —— | 57.5 | 34.0 | 59.9 | 29.1 |
| | IL+RL | BERT$^‡$ + BART$^‡$ | —— | 58.9 | 26.0 | 62.4 | 28.7 |
| | AgentQ$^†$ | xLAM-v0.1-r-46.7b$^‡$ | —— | —— | —— | —— | 50.5 |
| **Reward-Guided Inference** | Reflexion | gpt-3.5-turbo | —— | 77.2 | 46.0 | 72.9 | 41.3 |
| | LATS$^†$ | gpt-3.5-turbo | gpt-3.5-turbo | 75.9 | 38.0 | —— | —— |
| **Reward and Demo Free** | Greedy Baseline | gpt-3.5-turbo | llama-3.1-8b-instruct | 70.0 | 26.0 | 67.7 | 26.4 |
| | | gpt-3.5-turbo | r1-distill-llama-8b | 68.4 | 24.0 | 66.3 | 24.6 |
| | | gpt-3.5-turbo | gpt-3.5-turbo | 71.5 | 38.0*** | 70.6*** | 35.6*** |
| | | gpt-3.5-turbo | gpt-4o | 72.9* | 42.0*** | 71.5*** | 40.6*** |
| | | gpt-4o | llama-3.1-8b-instruct | 71.6 | 28.0 | 67.2 | 25.8 |
| | | gpt-4o | r1-distill-llama-8b | 71.6 | 32.0* | 66.5 | 25.6 |
| | | gpt-4o | gpt-3.5-turbo | 77.4*** | 46.0*** | 72.4*** | 38.8*** |
| | | gpt-4o | gpt-4o | 74.4** | 46.0*** | 71.4*** | **40.8***** |
| | MCTS Baseline | gpt-3.5-turbo | llama-3.1-8b-instruct | 71.9 | 34.0** | —— | —— |
| | Greedy w/ STL (**Ours**) | gpt-3.5-turbo | llama-3.1-8b-instruct$^‡$ | **78.3***** | 46.0*** | 72.8*** | 36.6*** |
| | | gpt-4o | llama-3.1-8b-instruct$^‡$ | 76.0*** | 40.0*** | **74.2***** | 40.6*** |
| **Human Expert** | —— | —— | —— | 76.1 | 54.0 | 82.1 | 59.6 |

# 4 Experiments

We benchmark our proposed STL self-improvement approach on applied web agent tasks, multi-step question answering, and math puzzle tasks[1].

## 4.1 Web Tasks

As mentioned in §1, it is particularly challenging and expensive to gather ground truth web task completion data [70]. To benchmark our STL method on web tasks, we utilize WebShop [63], which consists of interactive web tasks involving searching for and purchasing an item that matches a short natural language specification. This benchmark is an ideal test bed to demonstrate the ability of our approach, as, unlike other web task datasets [75, 31], ground truth reward is provided for all tasks, allowing a direct comparison between STL and methods in the Reward and Demo Learning and Reward-Guided Inference settings which use this reward.

**STL for web tasks.** Following the empirical advantages on agent tasks identified by [66], we generate training data with MCTS by performing a step of lookahead at each step during rollout. Note that we use the LLM value model value outputs as a *proxy reward* to guide UCT (Upper Confidence bounds applied to Trees) [30] selection like [66], instead of ground truth reward [74]. We perform STL with a gpt-3.5-turbo [5] policy to be consistent with previous work [74, 49] as well as a gpt-4o [43] policy and fine-tune a llama-3.1-8b-instruct [13] value function. STL is performed by rolling out 50 tasks from the WebShop training set, resulting in 1161 training examples, which we find is sufficient for significant performance improvement. We find that training a separate value model at each depth allows us to train with smaller LLMs with fewer active parameters during rollouts (see Appendix D.3 for more details). Also, while we perform data generation during STL with MCTS, we evaluate the agent using the trained value models with *greedy search*, where the next action is greedily chosen based on the values of the policy's proposed actions (see Algorithm 2 for more details). Finally, we find that a single iteration of STL is sufficient to see significant improvement over using a base LLM-initialized value model, and that multiple iterations do not yield additional performance improvements due to difficulties in simulating more than one step ahead, given the complexity of the environment. This conclusion has been corroborated by other LLM agent

---

[1]Our code is available at https://github.com/ethanm88/self-taught-lookahead.

works [20, 6]. However, in §4.3 we show that this multi-step simulation is possible for simpler tasks. See Appendix D for further details about data generation, training, and simulation.

**Baselines.**  Within  Reward and Demo Learning  approaches, we include IL and RL methods originally proposed in the WebShop work that train BERT [12] and BART [36] models on human demonstrations and ground truth reward [63]. We additionally compare to the current state-of-the-art approach AgentQ [47], which finetunes a larger xLAM-v0.1-r-46.7b policy on rolled out MCTS search trees using direct preference optimization [48]. We also include **Reward-Guided Inference** methods such as Reflexion [49] and LATS [74], which work by prompting closed-source LLMs within a framework, e.g., MCTS, that is guided by ground truth reward. Finally, in the  Reward and Demo Free  setting, we include greedy search and MCTS[2] baselines with a ReACT [65] prompted base LLM (llama-3.1-8b-instruct, gpt-3.5-turbo, gpt-4o, r1-distill-llama-8b [22]). All value models are prompted with few-shot examples and are asked to provide reasoning before a numerical value. Following LATS for a fair comparison, we use a branching factor of 5 for all methods and 30 iterations for MCTS-based approaches. This makes LATS and MCTS baselines strictly more computationally expensive than greedy methods. We use the pass@3 [8] for methods that do not have access to reward at inference time. Finally, we include a comparison to human expert performance measured in the original WebShop paper [63].

**Results and discussion.**  The WebShop average reward (Score) and success rate (SR) of evaluated methods are presented in Table 1, with additional baselines in Table 4. We present results on both the full WebShop test set and on the mini test set of 50 tasks used by [74], as we find running LATS and other MCTS methods on the entire test set is computationally expensive. Both of these sets are distinct from those seen during STL. Within the  **Reward and Demo Free**  setting, we find that STL matches the performance of using a gpt-4o value model and leads to a greater than 7% improvement in average reward and a 39% improvement in success rate relative to a base llama-3.1-8b-instruct value model, both of which are statistically significant improvements ($p < 0.001$) using the paired bootstrap test [3]. Moreover, we find that STL even performs similarly to  **Reward-Guided Inference**  methods that have access to ground truth rewards. Lastly, we find no statistically significant difference between using a gpt-3.5-turbo or gpt-4o policy due to high action diversity in our setup (see Appendix D.2); thus, we evaluate search methods with a single policy (either gpt-3.5-turbo or gpt-4o) in subsequent experiments.

**Reasoning ablation.**  In the  **Reward and Demo Free**  setting, we also perform ablations on the set of information from the step of lookahead used to fine-tune the value model during self-improvement. Specifically, we compare STL to variants that use only subsets of the information derived from lookahead. As mentioned in §3, this information includes **(1)** the lookahead value, **(2)** the textual representation of the next-best action and its successor state, and **(3)** the value rationale for the successor state. The results of this ablation in Table 2 demonstrate that regressing solely on lookahead values and further incorporating state transitions from lookahead does improve performance relative to the base model. However, learning also from the value rationale of the successor state, as done in STL, yields additional performance gains over these other settings. These results substantiate the claims made in §3 about the necessity of learning from action-outcome rationales, a key difference between STL and both classical RL [19] and other LLM tree

Table 2: Ablation study on the impact of fine-tuning with different combinations of information from lookahead, namely lookahead values (LV), textual representation of the next best action and successor state (TR), and the value rationale for the successor state (R). The underlined results are from the base model before any fine-tuning.

| Fine-tuning Data Setup | Score ↑ |
|---|---|
| llama-3.1-8b-instruct | 70.0 |
| + LV | 76.0 |
| + LV + TR | 74.4 |
| + LV + TR + R **(STL)** | **78.3** |

search works [16, 70] which fine-tune an LLM value model on numerical values only. Note for fairness, we modify the loss-masking to improve the lookahead value-only baseline and include this higher result in the table. See Appendix F.4 for more details about loss masking and this baseline.

---

[2] We use LLM value as a proxy reward to guide UCT, like in the data generation phase of STL

Table 3: Match rates on `HotpotQA`. Value functions marked with ‡ are fine-tuned. Statistical significance with the paired bootstrap test of **Reward and Demo Free** methods against the underlined results (***$p < 0.001$) is provided. The best **Reward and Demo Free** results are **bolded**.

| Setting | Method | Policy | Value | Match Rate ↑ | |
|---|---|---|---|---|---|
| | | | | Test Set (50) | Test Set (500) |
| **Reward and Demo Learning** | R1-Searcher | `llama-3.1-8b-instruct`‡ | — | 46.0 | 44.8 |
| **Reward-Guided Inference** | Reflexion | `gpt-3.5-turbo` | — | 70.0 | 66 |
| | LATS* | `gpt-3.5-turbo` | `gpt-3.5-turbo` | 70.0 | — |
| **Reward and Demo Free** | Greedy Baseline | `gpt-3.5-turbo` | `llama-3.1-8b-instruct` | 60.0 | 56.4 |
| | | `gpt-3.5-turbo` | `gpt-3.5-turbo` | 62.0 | 56.0 |
| | | `gpt-3.5-turbo` | `gpt-4o` | **68.0** | 57.6 |
| | Greedy w/ STL (**Ours**) | `gpt-3.5-turbo` | `llama-3.1-8b-instruct`‡ | 66.0 | **61.8**\*\*\* |

## 4.2 Multi-Hop Question Answering

We also investigate the efficacy of STL on applied reasoning for retrieval-based question-answering. We specifically utilize the `HotpotQA` [62] benchmark, consisting of multi-hop question-answering tasks that require retrieving and reasoning over multiple Wikipedia entries. We use the same setup as in §4.1 to generate data by rolling out with MCTS, but instead roll out 500 tasks from the training dataset since actions (search terms) proposed by the policy lack diversity compared to web tasks.

**Baselines.** In the **Reward and Demo Learning** setting, we evaluate on R1-Searcher [52], which performs RL on outcome-based answer correctness rewards on the multi-step retrieval QA task. As in §4.1, in the **Reward-Guided Inference** setting, we include Reflexion and LATS baselines, which have access to ground truth correctness during search. Among **Reward and Demo Free** methods, we compare STL to greedy search with closed-source models.

**Results and discussion.** Table 3 presents the answer match rate of all evaluated methods. Following [15], we prompt `gpt-4o` to help evaluate answer correctness. The prompt used is in Appendix E.3. This fuzzy matching correctness is exposed as the reward for Reflexion and LATS, but not to the greedy methods in the **Reward and Demo Free** setting. We evaluate on a set of 500 unseen questions and also a smaller set of 50 examples due to the high cost of LATS. STL on `llama-3.1-8b-instruct` outperforms a `gpt-4o` value model and the R1-Searcher method while approaching the performance of **Reward-Guided Inference** methods, which can verify predicted answer correctness during inference. Of the **Reward and Demo Free** approaches, STL is the *only method* that is statistically significant compared to the `llama-3.1-8b-instruct` baseline ($p < 0.001$).

## 4.3 Math Puzzles

Finally, we also study the performance of STL on the `Game-of-24` task [64], where the goal is to construct a mathematical expression with 4 provided integers to obtain 24. The task was originally for **Reward and Demo Free** methods, so it serves as a good benchmark for STL.

**STL for `Game-of-24`.** For this task, we generate data using breadth-first search (BFS) rather than MCTS to be consistent with the original Tree-of-Thoughts [64] approach. We use a `gpt-4o` policy and a `llama-3.1-8b-instruct` base value function in the first iteration. As described in §3, we replace this base value function with a trained model in each subsequent iteration. STL is run for four iterations of 25

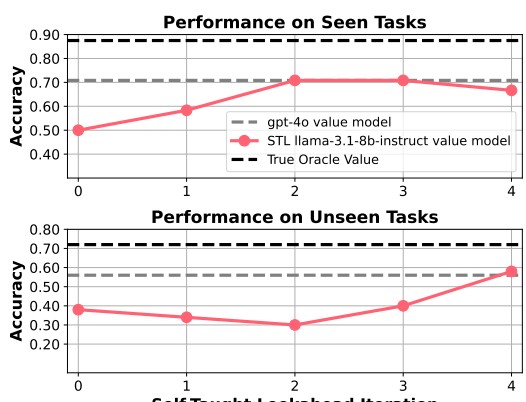

Figure 3: BFS `Game-of-24` performance on tasks seen and unseen during STL.

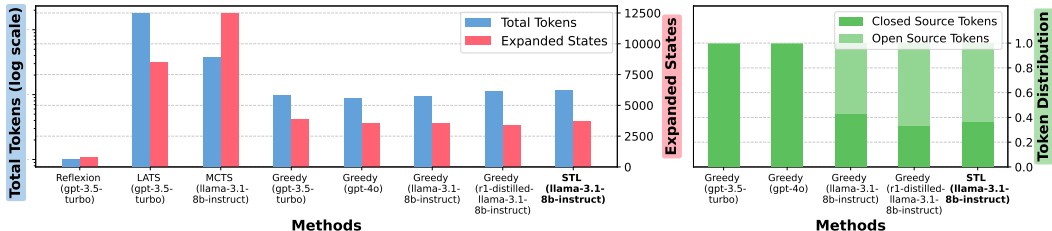

Figure 4: Compute and environmental efficiency during evaluation on `WebShop` with a `gpt-3.5-turbo` policy **(left)**. Compute efficiency is measured in total (prompt and completion) tokens. Environmental efficiency is measured by the number of states expanded (webpages visited). The distribution of tokens (closed vs. open source models) used during search is also shown **(right)**. Value models are specified in parentheses.

puzzles. For this task, we do not have explicit environment observations, but instead use the policy's arithmetic to combine two numbers as a *pseudo-observation*. See Appendix F for further details.

**Baselines.** We compare the performance of value models learned via STL with Reward and Demo Free BFS baselines that use the same `gpt-4o` policy. Specifically, we experiment with a `gpt-4o` value model and an algorithmic *oracle* evaluator [10]. This oracle runs a recursive algorithm to verify whether the current state (set of numbers) can be combined to reach 24. Search performance with this oracle is an upper bound on the performance improvement possible from improving the value function under a static policy.

**Results and discussion.** Figure 3 shows the performance of evaluated methods on a set of 50 tasks seen during STL and a set of 50 more challenging (determined by lower human solve percentages), unseen tasks. On both sets, STL matches or outperforms a `gpt-4o` value model. However, STL's performance on seen tasks monotonically increases for the first three iterations, while its performance on unseen tasks decreases before increasing. This phenomenon is due to the limited number of tasks the value model sees during training during the first couple of iterations. Specifically, if value models are not exposed to enough actions and their lookahead values during training, they fail to generalize well to unseen tasks. Limiting the number of tasks per iteration also limits the quantity and diversity of actions and values seen. A full analysis of this result can be found in Appendix F.5.

## 5 Efficiency Analysis

In this section, we compare the efficiency of STL with prior methods on `WebShop`. We study efficiency tradeoffs from two perspectives **(1)** model costs and **(2)** environment usage. Additionally, in §5.3 we explore how performance changes when scaling the size of the value model trained during STL.

### 5.1 Compute and Cost Efficiency

Keeping compute requirements and costs low is critical, especially for agents automating routine, repetitive tasks. Since STL can be used to improve an open-source LLM like `llama-3.1-8b-instruct`, we can transfer computation from more expensive closed-source models like `gpt-4o` to open-source alternatives while maintaining performance. Figure 4 **(right)** demonstrates this transfer, as STL uses more than 50% fewer tokens generated from closed-source than greedy search with a `gpt-3.5-turbo` or `gpt-4o` value model. We also compute the monetary costs of different methods. Unlike other methods, STL incurs costs for data generation and fine-tuning in addition to inference, but these are one-time costs that do not scale with agent use and are quite modest at $8.54 and $1.76, respectively. We plot inference costs against the average `WebShop` reward in Figure 5 **(left)**, and find that STL is Pareto optimal, $23\times$ cheaper than MCTS methods like LATS, and $5\times$ cheaper than performing greedy search with a `gpt-4o` value model. See Appendix G for details about the cost calculation.

### 5.2 Environmental Usage

It is often crucial for an agent taking actions in physical or digital environments to be *environmentally efficient* or conservative in the number of states it visits while performing a task. In the case of digital

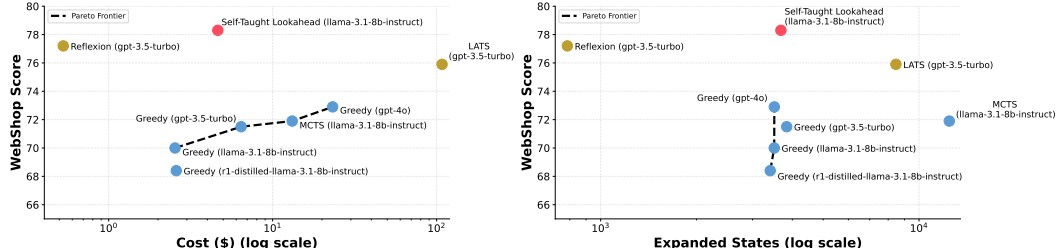

Figure 5: Tradeoff between performance and efficiency on `WebShop` with a `gpt-3.5-turbo` policy. Pareto frontiers of existing methods and baselines are shown, illustrating the optimality of STL when considering the tradeoff between inference cost and average reward **(left)** and between environmental usage and average reward **(right)**. **Reward-Guided Inference** methods are presented in gold and not included in the Pareto frontier since they belong to a different information setting.

web agents, taking many steps per task through exhaustive tree search may put an unnecessary burden on web servers, especially as agents are deployed at scale. Additionally, allowing web agents to search widely when equipped with personal information or the ability to make purchases may lead to unintended privacy disclosures or financial loss, respectively. In some environments, taking many actions may also lead to unreasonable task completion times. Figure 4 **(left)** and Figure 5 **(right)** present the environmental efficiency measured by the count of expanded states or visited sites in the `WebShop` environment. Considering `WebShop` score, STL is Pareto optimal and requires expanding half as many states as MCTS-based methods like LATS. Moreover, unlike LATS, STL does not require irreversible actions (actually clicking BUY NOW on a product page) required to obtain reward.

## 5.3 STL Scaling Trends

As 8B STL models can match the performance of a `gpt-4o` value model, is it possible to use *even smaller* models for STL while maintaining good performance? STL requires models to **(1)** provide generally consistent values out-of-the-box so that it is possible to compare successor states during data generation and **(2)** learn to generalize to unseen tasks and states, both of which may be challenging for smaller models. We explore this STL scaling trend on `WebShop` with ≤ 8 billion parameter models in the `llama` 3 family[3] [13] and ≤ 7 billion parameter models in the `qwen-2.5-instruct` family [61]. The results presented in Figure 6 demonstrate that

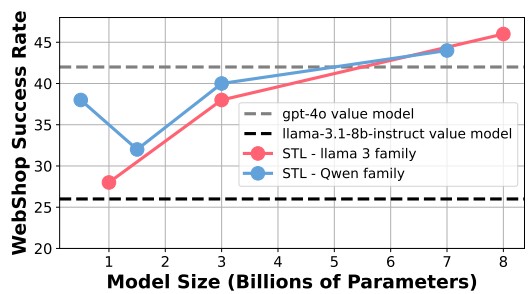

Figure 6: STL scaling trends on `WebShop` for `llama-3` and `qwen-2.5` families with a `gpt-3.5-turbo` policy.

while performance does generally decrease with fewer parameters, the smaller 3B parameter models in both families do approach `gpt-4o` performance. This result suggests that using smaller models for STL is feasible, making large-scale agent deployment to new domains more practical.

## 6 Discussion: When to use STL?

As discussed in §4 and further detailed in Appendix D.2, achieving strong performance with STL requires high action diversity (a large number of possible actions at each step). When action diversity is low, the effect of the value model on search performance is diminished, and it is necessary to roll out more tasks to obtain enough data to fine-tune the value model. For example, Section 4.2 notes that due to its low action diversity, HotpotQA shows smaller relative gains from STL compared to WebShop and also requires rollouts of 500 tasks (10 times more than WebShop) to get a large enough dataset for fine-tuning.

Additionally, STL requires later states to provide good value estimates that can be backed up and learned during training on lookahead results. Search performance will benefit most from STL on

---

[3]As the `llama-3.1` family lacks smaller models, we use 1B and 3B models from the `llama-3.2` family.

tasks where state transitions are consistent throughout the environment, i.e., the same or semantically similar actions yield similar outcomes. For instance, clicking the "Low to High" button on any search results page consistently orders items by price. This consistency enables the STL value model to accurately simulate a step of lookahead (§3.3), leading to better state-value estimation and improved downstream search performance. Tasks that have stochastic transitions or have inconsistent transitions when actions are semantically similar may not show large improvements with STL. Fortunately, most popular tasks where LLM agents have been deployed, such as web navigation, have deterministic and fairly consistent transitions.

# 7    Related Work

**Classical RL.**    STL is loosely inspired by fitted value iteration (FVI) [19], which generalized value iteration [2] to the tabular setting. In an iteration of FVI, target values are computed using the Bellman update and used to train a new value model from the previous model checkpoint using least squares regression. The iterated values in FVI are computed similarly to the lookahead values $y_{s_k}$ in §3.1, but with STL, no ground truth reward is assumed, the value model is non-Markovian, and actions are deterministic. Instead of learning directly from iterated values, with STL, they are concatenated with action-outcome rationales, and together, these sequences are used to fine-tune the LLM value model from scratch at each iteration rather than from the previous model checkpoint as in FVI.

**LLM self-improvement.**    A variety of previous work has shown that LLMs can self-improve with iterative prompting techniques [26, 58, 39] and have applied these methods to various domains, from agents [29] to privacy protection [9]. A separate line of work focuses on bootstrapping a small training dataset through a *self-training* process to improve either the reasoning policy model [21, 51, 28] or the verification or reward model [24] using synthetically generated data. While most self-training approaches utilize outcome-based reward models, other work [1, 70] derive process-based rewards like STL to evaluate each step in the reasoning chain.

**Training reasoning agents.**    The majority of prior work on training reasoning agents focuses on performing SFT on human-annotated trajectories [63, 34], synthetically generated trajectories [7, 17, 72, 38, 45, 41, 44, 53], or a combination of both [69, 71]. Other work has trained agents from tree search-generated data [18, 47, 70] or with explicit reinforcement learning  [46, 57, 14], but these methods usually require ground truth reward. While prior work has explored self-improving reasoning agents, these approaches fail to generalize beyond the instructions encountered during self-improvement [45] or require fine-tuning frontier models like `gpt-4o` to achieve generalization [66].

# 8    Conclusion

We propose STL as an efficient method to improve the value model employed during search. This efficiency primarily stems from STL's design for information-scarce settings, where models learn from state-transition dynamics. Additionally, because STL enables self-improvement on small models deployable with less exhaustive search, it yields significant reductions in both computational cost and environmental usage. Therefore, the STL framework could help enable the more realistic learning and deployment of agent systems.

## Acknowledgments

We would like to thank Microsoft's Azure Accelerate Foundation Models Research Program and NVIDIA's Academic Grant Program for providing computational resources to support this work. This research is supported in part by the NSF under grant number IIS-2052498 and SMA-2418946. Any opinions, findings, and conclusions or recommendations expressed in this material are those of the author(s) and do not necessarily reflect the views of the National Science Foundation. We also appreciate Jungsoo Park, Yao Dou, Geyang Guo, and the anonymous NeurIPS reviewers for their valuable feedback, which helped to improve the paper.

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

---

**Algorithm 1** Self-Taught Lookahead

---
**Require:** Set of tasks $\mathcal{D}_{\texttt{rollout}}$, Base LLM $M$, num iterations $n$, num tasks per iteration $m$

1:   $\pi_\theta \leftarrow \texttt{initialize\_policy\_model}(M)$
2:   $V_{\phi_0} \leftarrow \texttt{initialize\_value\_model}(M)$
3:   $\mathcal{D}_{\texttt{Train}_0} \leftarrow \{\}$
4:   **for** $k \leftarrow 1$ to $n$ **do**
5:      $\mathcal{D}_{\texttt{rollout}_k} \leftarrow \mathcal{D}_{\texttt{rollout}}[m \cdot (k-1) : m \cdot k]$           $\triangleright$ Select tasks for iteration $k$
6:      **for** $x \in \mathcal{D}_{\texttt{rollout}_k}$ **do**
7:          $T \leftarrow \texttt{rollout\_tree}(\pi_\theta, V_{\phi_{k-1}}, x)$        $\triangleright$ Generate rollout search tree for task $x$
8:          $y \leftarrow \texttt{calculate\_lookahead\_values}(T)$
9:          $o \leftarrow \texttt{calculate\_action\_outcomes\_rationales}(T)$
10:         $y_{\texttt{filtered}}, o_{\texttt{filtered}} \leftarrow \texttt{task\_specific\_filter}(y, o)$     $\triangleright$ Apply task-specific filtering if applicable
11:         $\mathcal{D}_{\texttt{Train}_k} \leftarrow \texttt{add\_new\_data}(\mathcal{D}_{\texttt{Train}_k}, (y_{\texttt{filtered}}, o_{\texttt{filtered}}))$
12:      **end for**
13:      $V_{\phi_k} \leftarrow \texttt{fine\_tune}(V_{\phi_0}, \mathcal{D}_{\texttt{Train}_k})$             $\triangleright$ Finetune from base model
14:   **end for**

---

## A   Broader Impacts

Our STL method enables LLMs to self-improve at search by capturing the mechanics of traditional RL algorithms in natural language. While we leverage this technique for better state-value estimation, it can likely be applied to other problems. Additionally, we show in §5, that STL requires significantly less compute overhead than similarly performing methods, thus reducing the energy consumption required for search during inference, helping to improve the sustainability of agent deployment.

As with all self-improvement work [45, 66], having models self-improve without human supervision may enable agents to learn how to take actions towards task completion that are not well-aligned with human values or preferences. To prevent misuse, we only release code to reproduce experiments in this paper and not to improve general-purpose agents for which these risks might be more prevalent. While these harms are out of the scope of this work, we encourage future research in this area.

Finally, we note that it is possible to use our value model as a sort of policy to directly select actions in environments with a finite set of actions from each state. While we did not explore how well such a policy would work, improving value estimation to improve the policy may be an interesting direction for future work.

## B   Limitations

During self-improvement, STL does require task specifications like "I am looking for a queen-sized bed that is black ..." (see Figure 2) for web tasks. However, this assumption is well-founded as prior self-improvement work, such as that on LLM alignment [67, 59], also assumes tasks, or in their case, user prompts, are provided to initialize the self-improvement process. Additionally, as mentioned throughout the paper, this setting is much more reasonable than **Reward and Demo Learning** and **Reward-Guided Inference** settings, which require ground truth reward and/or human demonstrations along with task specifications.

Additionally, due to compute constraints, our experiments utilize models up to 8 billion parameters for STL. However, our results in §5 demonstrate a scaling trend in performance with respect to model size, indicating that using larger models may yield more performant agents. Given these positive results, we leave applying STL to larger models to teams with larger resource budgets.

STL requires actions to be grounded in language to be able to leverage LLMs' strong priors. Therefore, STL may struggle in traditional RL tasks like gridworlds, which may have a large state and action space and transitions that are not grounded in language. However, prior work has contextualized gridworld environments by adding landmarks [11], so similar techniques may be used to obtain the necessary consistent and grounded transitions for STL to be successful.

**Algorithm 2** Greedy Search

---

**Require:** LLM policy $\pi_\theta(\cdot)$, LLM value model $V_\phi(\cdot)$, Initial state $s_o$
1:   $s_i \leftarrow s_0$
2:   **while** $s_i$ is not terminal **do**
3:      $a_i \leftarrow \arg\max_{a \in A_{s_i}} \left\{ \text{Val}(V_\phi(x, s_0, \ldots, T(s_i, a)) \right\}$         ▷ Greedily pick best action
4:      $s_{i+1} \leftarrow T(s_i, a_i)$
5:      $i \leftarrow i + 1$
6:   **end while**

---

## C    Algorithms

The STL algorithm is presented in full in Algorithm 1. For information about the `task_specific_filter`, see Appendix D.3 and Appendix F.4.

Additionally, the algorithm for greedy search as used in §4 is presented in Algorithm 2.

## D    Self-Taught Lookahead on `WebShop`

In this section, we outline information about `WebShop` [63] and the implementation details of running STL on the benchmark.

### D.1   `WebShop` Task

There are two main types of actions in the `WebShop` task:

- `search[query]`:
  Search actions allow the user to search for a particular item with a natural language query, e.g., `search[easy to use medium color face kit less than 40 dollars]`. This action can only be taken on the search page, which is also the initial / home page of the `WebShop` interface.

- `click[button]`: Click actions are discrete actions but can take many forms, which we enumerate below:
  - `click[product]`: to select a relevant product from the search results e.g. `click[B09B6SH764]` where B09B6SH764 is a product code.
  - `click[attribute]`: to toggle on an attribute or option on the product page of an item, e.g. `click[small]`
  - `click[Buy Now]`: to buy the selected item - this is a terminal action that yields the ground truth reward. This action is not allowed to be taken in STL search but is allowed in other search, RL, and prompting methods [63, 49, 74].
  - Other navigation buttons: other navigation buttons include `click[Back to Search]`, `click[<Prev]`, `click[Next>]`, `click[Description]`, `click[Features]`. To simplify trajectories, we generally restrict the ability for models to take these actions in all settings following [63].

`WebShop` provides a textual representation of webpages in `simple mode`. An example of this representation for search results is shown in Figure 7.

### D.2   Prompts

The prompt used to generate actions in `WebShop` is presented in Figure 8. Notice that we do not use `think` actions part of the classical ReACT framework [65] like [63] or [74] because evaluating the value of these actions is difficult as they have no observation. Instead, we prompt the policy model to provide a rationale while generating possible actions. Also, note that we prompt the policy multiple times, adding to the list of actions that are not allowed and removing from the list of actions that are allowed. This change to a "selection" policy enables action diversity, which we find is otherwise low

```
┌─ WebShop Textual Representation ─────────────────────────┐
│                                                          │
│ [Back to Search] Page 1 (Total results: 50)             │
│ [Next >]                                                 │
│ [B0972Q1T8T]                                             │
│ Cosycost USB Microphone, Condenser Computer PC Gaming    │
│ Microphone for PS4/5 Laptop Windows Mac OS Android       │
│ Phone, Noise Cancelling Instant Mute, Studio Mic for     │
│ Voice, Music Recording, Podcasting, Streaming            │
│ $32.99                                                   │
│ [B09N3M6H2Z]                                             │
│ Wired Stereo Headset Noise Cancelling Microphone with    │
│ in-line Controls/Volume Controller, All-Day Comfort      │
│ Design, Works for Playstation, Nintendo Switch, PC       │
│ with USB Connection (HS-HP101UNCBK)                      │
│ $199.99                                                  │
│ [B072L2D6LY]                                             │
│ Andrea Communications NC-255VM USB On-Ear Stereo USB     │
│ Computer Headset with Noise-Canceling Microphone,        │
│ in-Line Volume/Mute Controls, and Plug                   │
│ $34.59                                                   │
│ [B071H84LTJ]                                             │
│ Andrea Communications NC-455VM USB Over-Ear Circumaural  │
│ Stereo USB Computer Headset with Noise-Canceling         │
│ Microphone, in-Line Volume/Mute Controls, and Plug       │
│ $49.24                                                   │
│ [B08GLJSWJ9]                                             │
│ Jiade USB Headset with Noise Canceling Microphone for    │
│ CallCenter Skype Chat, Computer Phone Headset Voice      │
│ Recognition Speech Dictation, PC Headphone with Mic      │
│ Mute Volume Control Binaural Golden                      │
│ $9.99                                                    │
│                                                          │
└──────────────────────────────────────────────────────────┘
```

Figure 7: Example of `simple` mode textual representation of the state with the `WebShop` benchmark.

even with prompting the policy at high temperature. This change also likely explains why there is no statistically significant difference between using a `gpt-3.5-turbo` and `gpt-4o` policy in §4.1.

Likewise, the prompt used to evaluate states is presented in Figure 9. Note that this evaluation prompt is only used to prompt base models; STL value models are only prompted with the current trajectory. We note that the Likert scale used was crucial to obtaining consistent value outputs on which we could perform STL. We also use a special value estimation prompt whenever on a product listing page to select attributes (see Figure 10) in order to obtain consistent values. Specifically, we convert the 4-point scale into a -2 to 2 scale and add the value to the value of the prior state's (product selection action) value. We tried to use this in our baselines, but we found that it actually harmed the baseline scores. However, in exploratory experiments, it helped yield more consistent values, so we use this for STL.

For all value estimates (base model or fine-tuned), we prompt the value model 5 times and use the average score as the state value estimate. During the data generation phase, since we need a single rationale to fine-tune on which to construct the action-outcome rationale, we choose the rationale corresponding to the median of the 5 scores. After the rebuttal period, we found an instruction to further discount values in the prompts to the fine-tuned models. While we could not rerun all results due to inference costs, we reran the STL evaluation on the mini test set and found results were maintained.

In total, the data generation phase on `WebShop` yielded a total of 1162 examples, collected from rolling out search trees for 50 tasks. An example of the rationale structure of the training data is presented in Figure 11.

Figure 8: Generation prompt for WebShop policy.

## D.3 Implementing STL

With web tasks, the position of an action in a trajectory may influence its value. For instance, selecting a certain item *I* from search results early in the trajectory should have a higher value than selecting *I* a second time in the same trajectory. To account for this difference, we train value models at each position (depth) in the trajectory. Specifically, we limit trajectories to five steps and train four value models depths 1 to 4, only allowing a terminating BUY action on the final step.

We also filter out malformed rationales from the training data. Specifically, we remove rationales that do not provide the proper format, e.g., it does not exactly contain scaffolding like "Thus the correctness score is".

Additionally, we generate lookahead rollouts using 5 iterations of MCTS, which, in practice, we find is sufficient for collecting diverse training data.

Finally, we compute loss over both input and output tokens to enable the value model to more quickly capture the dynamics of the environment from the trajectory. However, as we mentioned in §4.1, this practice harms the performance in the setting where the model regresses only on the lookahead value. Therefore, we found that when we only compute loss on the output tokens as in the normal SFT setting, the WebShop score in this setting increases from 70.9 to 76.0. This score is actually higher than the setting where we predict the future next best state without a rationale, which indicates that rationales may be key when we try to learn from transition dynamics. We are not sure why this is the case, and why there is not a larger gap between STL and this lookahead value baseline, but we believe that it could be due to small differences in the prompt due to different information settings and/or the prediction or future state potentially harming value estimation for clearly good or bad states.

## D.4 Difficulty in Performing STL for Multiple Iterations.

Empirically, we find that STL after a second iteration on WebShop has a lower performance (average reward of 68.6, and success rate of 26.0) than after a single iteration. From a manual inspection of the lookahead and rationales generated, we notice that the second step simulated by the value model often does not match the true environment.

> **WebShop Value Estimation Prompt**
>
> Given an item to purchase and a trajectory that aims to buy an item that exactly matches the specification, analyze how well the last action and observation align with the task. Provide a reflection that concludes with. "Thus the correctness score is s", where s is either 1, 2, 4, 6, 8, or 10. Use the following scale for scoring:
> 1: The last action and observed state is entirely irrelevant to the task or captures a purchase of an item that is completely unrelated to the specifications.
> 2: The last action and observed state captures a step with a low likelihood of leading to purchasing the correct item.
> 4: The last action and observed state captures a step with a moderate likelihood of leading to purchasing the correct item.
> 6: The last action and observed state captures a step with a high likelihood of leading to purchasing the correct item.
> 8: The last action and observed state captures a step with a very high likelihood of leading to purchasing the correct item.
> 10: The last action and observed state captures a step that will certainly lead to purchasing the correct item.
>
> Keep reflections short (<100 words). Follow the format of the rationale from the below example task.
> NOTE: the observation from clicking on the item will be the item's product detail page. For instance, click[B078GWRC1J] will show the product detail page for the item with code B078GWRC1J which will include the item's name (e.g. Bright Citrus Deodorant by Earth Mama), price ($10.99), and other relevant details as well as options.
> NOTE: Assume none of the attributes on the product page are selected only provide the reflection for the last action.
> Example Tasks:
> {few shot examples}
>
> ---
>
> New Task:
> Respond with the reflection for the last observation of the new task ONLY. As a reminder the last action and observation is as follows: {last_action} Your response should start with "Reflection:" and end with "Thus the correctness score is ...".

Figure 9: Value Estimation prompt for WebShop. This prompt was only used to prompt base models.

Table 4: Additional baselines on WebShop. See Table 1 for the full results.

| Setting | Method | Policy | Value | Mini Test Set (50) | |
|---|---|---|---|---|---|
| | | | | Score ↑ | SR ↑ |
| **Reward and Demo Free** | ReACT (No Search) | gpt-3.5-turbo | — | 68.9 | 36.0 |
| | | gpt-3.5-turbo | — | 70.0 | 36.0 |
| | Greedy Baseline | gpt-3.5-turbo | qwen3-8b | 68.6 | 34.0 |

## D.5 Additional Results on WebShop

We have included performance on other baselines in Table 4, including ReACT [65] baselines with OpenAI models and an additional greedy search baseline with a qwen3-8b reasoning value model. Results with error bars are also provided in Table 5.

```
┌─ WebShop Attribute Prompt ──────────────────────────────────┐
```

Given an item to purchase and a trajectory that aims to buy an item that exactly matches the specification, analyze how well the last action and observation align with the task. All the last actions you see will be selecting an attribute on the product page of a candidate item. Provide a reflection that concludes with "Thus the correctness score is s", where s is either 1, 2, 3, or 4. Use the following scale for scoring:

1: The attribute selected is opposite to the specified attribute in the task.
2: The attribute selected is irrelevant to the specified attribute in the task.
3: The attribute selected is an attribute mentioned in the instruction, but not all attributes mentioned are currently selected.
4: The attribute selected is an attribute mentioned in the instruction, and all attributes mentioned are currently selected.

Keep reflections short (< 100 words). Follow the format of the rationale from the example task below.
NOTE: When selecting attributes like click[x-large], click[red], etc., the observations will be "You have clicked x-large" or "You have clicked red" respectively.
IMPORTANT: Assume options for size, weight, color, etc. on product pages are not selected unless explicitly mentioned in the trajectory (e.g., click[large], click[red], etc.). Selecting these attributes may be crucial to match the task requirements. Only provide the reflection for the last action.

**Example Tasks:**
{few shot examples}

─────────────────────────────────────────────────

**New Task:**
{input}
Respond with the reflection for the last observation of the new task ONLY. As a reminder, the last action and observation is as follows:
{last_action}

Your response should start with "Reflection:" and end with "Thus the correctness score is ..."

Figure 10: Value estimation attribute prompt for `WebShop`. This prompt was only used to prompt base models during rollouts.

## D.6    Reflexion Performance on `WebShop`

**Tracing bugs in Reflexion implementation**    We originally ran Reflexion baselines on WebShop using the official Reflexion GitHub repository[4] with three iterations, changing only the model from `text-davinci-003` (a deprecated text completion model used in the original paper) to gpt-3.5-turbo. During the author rebuttal period, we found the following:

- Other researchers had obtained similar Webshop performance to our initial Reflexion results ( 15%) using the unchanged official implementation with gpt-3.5-turbo (see GitHub issue #49).

- Others also noted that changes to the prompts were needed to adapt the framework to conversational models to see improved performance in the 30-40% range (see GitHub issue #48).

- Another researcher also identified a bug in the WebShop implementation that prevented the use of memory from prior iterations (see GitHub issue #36). In the discussion of this issue, the first author of the Reflexion paper acknowledged that this bug may have caused the lack

---

[4]https://github.com/noahshinn/reflexion

┌─ WebShop Rationale Structure ─────────────────────────────────┐

**Task and Current Trajectory:**
Instruction: i'm looking for a pair of women's workout shorts with a drawstring waist. i need them to be extra large and in light gray, and price lower than 60.00 dollars [Search]
Action: search[women's workout shorts drawstring waist extra large light gray < 60 dollars]
Observation: *removed for brevity*
Current Action: click[B09T3PJM1R]
Current Observation: *removed for brevity*

**STL Output:**
I will evaluate the best successor state from the current state:
Best Next Action: click[x02c-gray]
Observation of Best Successor State: You have clicked x02c-gray.
Reflection of the Best Successor State: The last action selects the color 'x02c-gray' for the item B09T3PJM1R. Based on the observation, this product's color is indeed gray, which matches the specified criteria. Therefore, this product matches one of the attributes mentioned in the task. The last action and observation thus capture a step that selects an attribute mentioned in the instruction, but not all attributes mentioned (specifically the size attribute) are currently selected. Thus, the correctness score is 6.00 / 10.00.

Figure 11: Example of rationale structure from the STL training data rolled out on the WebShop task.

Table 5: Score and success rate (SR) on WebShop with error bars. The full results are in Table 1.

| Method | Policy | Value | Full Test Set (500) | |
|---|---|---|---|---|
| | | | Score ↑ | SR ↑ |
| Greedy Baseline | gpt-3.5-turbo | llama-3.1-8b-instruct | 67.7 ± 2.26 | 26.4 ± 3.86 |
| | gpt-3.5-turbo | r1-distill-llama-8b | 66.3 ± 2.30 | 24.6 ± 3.78 |
| | gpt-3.5-turbo | gpt-3.5-turbo | 70.6 ± 2.43 | 35.6 ± 4.23 |
| | gpt-3.5-turbo | gpt-4o | 71.5 ± 2.51 | 40.6 ± 4.33 |
| | gpt-4o | llama-3.1-8b-instruct | 67.2 ± 2.30 | 25.8 ± 3.84 |
| | gpt-4o | r1-distill-llama-8b | 66.5 ± 2.33 | 25.6 ± 3.83 |
| | gpt-4o | gpt-3.5-turbo | 72.4 ± 2.40 | 38.8 ± 4.29 |
| | gpt-4o | gpt-4o | 71.4 ± 2.49 | 40.8 ± 4.35 |
| Greedy w/ STL (**Ours**) | gpt-3.5-turbo | llama-3.1-8b-instruct[‡] | 72.8 ± 2.32 | 36.6 ± 4.22 |
| | gpt-4o | llama-3.1-8b-instruct[‡] | **74.2 ± 2.38** | **40.6 ± 4.30** |
| **Human Expert** | —— | —— | 82.1 | 59.6 |

of improvement of the Reflexion agent on WebShop that was reported in Appendix B.1 of the original Reflexion paper [49].

After modifying the prompts and patching this memory bug, the ReAct success rate on the 50-task test set is 36%, and the Reflexion success rate is 46% after three iterations, which is similar to the performance reported in other work [73]. We hope that this debugging process can be of some use to the community.

**Reflexion vs. LATS.** In Table 1, Reflexion outperforms LATS, which is potentially unexpected as LATS is more computationally expensive than Reflexion. However, due to differences in the mechanisms of these two methods, it is not necessarily the case that LATS performance is lower-bounded by Reflexion performance. For instance, Reflexion enables the LLM policy to make wholesale changes to any steps in its trajectory after reflecting on previous failures. On the other hand, while LATS provides its policy and value LLMs with reflections on previous trajectories, it still relies on the traditional (non-neural) mechanics of MCTS. For example, LLM reflection is not involved during node selection, and instead, the traditional Upper Confidence bounds applied to Trees (UCT) heuristic is used.

# E  Self-Taught Lookahead on `HotpotQA`

## E.1  HotpotQA Task

The `HotpotQA` [62] is a multi-hop question answering benchmark where correct answers require reasoning over multiple Wikipedia entries. There are three possible actions at each step:

- `search[entry]`:
  Search provides the first five sentences of the corresponding Wikipedia entry if it exists, or provides five alternative existing entries.
- `lookup[string]`:
  Lookup returns the next sentence in the entry containing the specified string.
- `finish[string]`:
  Finish signifies the completion of the reasoning process, where the answer is specified with the provided string.

## E.2  Prompts

The prompt used to generate actions for the `HotpotQA` task is presented in Figure 14. Likewise, the prompt used to evaluate states is presented in Figure 13. Many of the prompting details, such as the use of a disallowed action list, are similar to the `WebShop` task. However, one difference is that if five unique actions are not found due to a search term not being an entry, we add the top-5 similar terms returned by the retrieval model as possible actions.

In total, the data generation phase on `HotpotQA` yielded a total of 2708 examples, collected from rolling out search trees for 500 tasks.

## E.3  Evaluation

We use the same `gpt-4o` evaluation prompt as [15] which is provided in Figure 13. Note that a match is computed when either there is an exact match or a match indicated by the `gpt-4o` evaluator in order to prevent any mistakes by the evaluator, such as indicating a non-match even though the two answers were an exact string match.

Additionally, unlike `WebShop`, we limit trajectories to a depth of four, and explicitly prompt models to provide a final answer on the fourth step in the trajectory.

## E.4  Implementing STL

The implementation details of STL for `HotpotQA` are similar to `WebShop` in the training of multiple models at different depths, the filtering out of malformed rationales, and the use of 5 iterations of MCTS during data generation. However, one difference is that we feed the possible next actions to the value model so that it can provide coherent lookahead simulations and evaluations.

# F  Self-Taught Lookahead on `Game-of-24`

## F.1  `Game-of-24` Task

As introduced by [64], the `Game-of-24` is a mathematical reasoning task that involves combining four numbers e.g. "2 3 4 5" together with mathematical operations i.e. $+, -, /, \times$ in order to obtain 24. An action in this task consists of simply applying a mathematical operation to combine two numbers e.g. $2 + 3 = 5$, the resulting state from the operation is the set of remaining numbers e.g "5 4 5".

## F.2  Prompts

The prompt used to generate actions for the `Game-of-24` task is presented in Figure 15. Likewise, the prompt used to evaluate states is presented in Figure 16. Like `WebShop`, this evaluation prompt is only used to prompt base models; STL value models are only prompted with the current trajectory. However, unlike `WebShop`, values are not real numbers between 1 and 10, but rather 0.001, 1, and

```
┌─ HotpotQA Generation Prompt ──────────────────────────────────┐
│                                                               │
│  Solve a question answering task with interleaving Thought,   │
│  Action, Observation steps. Thought can reason about the      │
│  current situation, and Action can be three types:            │
│                                                               │
│  (1) Search[entity], which searches the exact entity on       │
│  Wikipedia and returns the first paragraph if it exists. If   │
│  not, it will return some similar entities to search.         │
│  (2) Lookup[keyword], which returns the next sentence         │
│  containing keyword in the current passage.                   │
│  (3) Finish[answer], which returns the answer and finishes    │
│  the task.                                                    │
│                                                               │
│  After each observation, provide the next Thought and next    │
│  Action.                                                      │
│                                                               │
│  NOTE: You MAY NOT select actions in the Actions Not Allowed  │
│  list. You have to change the wording of the query or lookup  │
│  somehow.                                                     │
│                                                               │
│  NOTE: Keep search queries and lookups short and concise      │
│  since longer queries will not return any result. For         │
│  instance, instead of searching Search[eastern sector of      │
│  the Colorado orogeny], search Search[Colorado orogeny] and   │
│  then Lookup[eastern sector].                                 │
│                                                               │
│  Here are some examples: {few shot examples}                  │
│  ───────────────────────────────────────────────────────     │
│                                                               │
│  New Task: {task}                                             │
│  Actions Not Allowed: {not_allowed_actions}                   │
│  Thought:                                                     │
│                                                               │
└───────────────────────────────────────────────────────────────┘
```

Figure 12: Generation prompt for the `HotpotQA` policy.

20, corresponding to the labels of impossible, likely, and sure that the remaining numbers can be combined to reach 24. Note that these values were used by the original Tree-of-Thoughts paper [64], but are ad-hoc and are used purely as labels. For all value estimates (base model or fine-tuned), we prompt the value model 3 times and use the median score as the state value estimate. During the data generation phase, since we need a single rationale to fine-tune on which to construct the action-outcome rationale, we choose the rationale corresponding to the median of the 3 scores.

### F.3 Evaluation

For all tested BFS methods, we use the same setup as the Tree-of-Thoughts paper, i.e., we select 5 of the best actions (set the branching factor to 5) at each of the 4 steps (two numbers are combined during each step).

### F.4 Implementing STL

Since the state space is quite limited, we combine training examples from the previous $k-1$ iterations with current examples to train the value model in the $k^{\text{th}}$ iteration. If the same state is encountered multiple times in different iterations, we defer to the value judgment from the latest iteration.

Unlike with `WebShop` and `HotpotQA`, we do not train value models at each depth, due to the small state space. Additionally, we also use a branching factor of 5 during data generation.

Table 6: Performance comparison across different numbers of tasks seen during self-improvement.

| Tasks Seen During Improvement | Accuracy (25 tasks / iter) | Accuracy (50 tasks / iter) |
|---|---|---|
| 0 | 38.0 | 38.0 |
| 25 | 34.0 | - |
| 50 | 30.0 | **48.0** |

```
┌─ HotpotQA Value Estimation Prompt ──────────────────────────────┐
│                                                                 │
│  Given a question and a trajectory to answer the question,      │
│  analyze how well the LAST ACTION in the trajectory contributes │
│  to finding the answer. Consider ONLY the last action.          │
│                                                                 │
│  The trajectories are labeled by pairs of thoughts that can     │
│  reason about the current situation and actions that can be     │
│  of three types:                                                │
│                                                                 │
│  (1) Search[entity], which searches the exact entity on         │
│  Wikipedia and returns the first paragraph if it exists. If     │
│  not, it will return some similar entities to search.           │
│  (2) Lookup[keyword], which returns the next sentence           │
│  containing keyword in the current passage.                     │
│  (3) Finish[answer], which returns the answer and finishes      │
│  the task.                                                      │
│                                                                 │
│  Provide a reflection that concludes with "Thus the             │
│  correctness score is s", where s is either 1, 3, 5, 7, or 10.  │
│  Use the following scale for scoring:                           │
│                                                                 │
│  1: The action is completely irrelevant to answering the        │
│  question or there is no relevant search result ("Could not     │
│  find" is in the observation).                                  │
│  3: The action's observation provides information only at the   │
│  background level to answering the question.                    │
│  5: The action's observation provides information that makes a  │
│  small step towards answering the question.                     │
│  7: The action's observation provides information that makes a  │
│  key step towards answering the question.                       │
│  10: The action's observation provides the final piece of       │
│  information needed to answer the question.                     │
│                                                                 │
│  Reminder: If you see "Could not find" in the observation, the  │
│  correctness score is 1.                                        │
│                                                                 │
│  Keep reflections short (< 100 words).                          │
│                                                                 │
│  Follow the following examples:                                 │
│  {few shot examples}                                            │
│  ─────────────────────────────────────────────────────────     │
│  {input}                                                        │
│                                                                 │
└─────────────────────────────────────────────────────────────────┘
```

Figure 13: Value estimation prompt for HotpotQA. This prompt was only used to prompt base models.

## F.5 Investigating Improvement Dynamics

In §4.3, we claimed that the initial decrease in performance on unseen tasks in Game-of-24 was due to a lack of sufficient tasks seen during self-improvement. We confirmed this claim by rerunning the experiment by rolling out 50 instead of 25 tasks per iteration. The results are summarized in Table 6, from which we see that performance actually improves in the first iteration if the number of tasks seen during self-improvement is increased to 50.

# G  Costs

Here we detail how we computed costs in the efficiency analysis in §5.

> **`HotpotQA` Evaluation Prompt**
>
> Based on the provided question and reference answer, please determine if the
> response is correct or incorrect. Begin by articulating your rationale, and
> conclude with a single word judgment: 'Yes' for correct or 'No' for incorrect.
> question: {question}
> reference answer: {reference}
> response: {response}

Figure 14: Evalaution prompt for the `HotpotQA` predicted answers.

> **`Game-of-24` Generation Prompt**
>
> Use numbers and basic arithmetic operations (+ - * /) to obtain 24. In each step, you
> are only allowed to choose two of the remaining numbers to obtain a new number.
> Follow the example format exactly.
> {few shot examples}
> ——————————————————————————————————————————–
> {input}

Figure 15: Generation prompt for the `Game-of-24` policy.

## G.1 Data Generation Costs

To compute data generation costs during STL, we use OpenAI's [5] for `gpt-3.5-turbo` and Groq's [6] pricing tables for closed source `llama` models. Note that we only use Groq for a pricing estimate, as inference and training were run on in-house GPUs. As costs may change over time, we provide the pricing figures that we used for our experiments in Table 7. We also note that experiments were initially run on `gpt-3.5-turbo-0613` on OpenAI Azure, which is multiple times as expensive as the newer `gpt-3.5-turbo-0125` due to resource allocation on Azure, despite being the better model. We therefore choose to report the data generation costs using the newer pricing point as it is more comparable to the current prices of other models.

## G.2 Fine-tuning Costs

We account for the costs incurred from fine-tuning `llama-3.1-8b-instruct` during STL. Fine-tuning with a single A40 GPU takes 4.5 hours for the WebShop task. While it is difficult to estimate costs on our own in-house GPUs, we can estimate the cost by using VastAI's figure of \$0.39 per A40 GPU hour [7] when these experiments were run in April, 2025. Thus, the total training run cost roughly \$1.76. We also note that as the number of tasks seen at test time scales, the fine-tuning costs become increasingly negligible.

## G.3 Inference Costs

To compute inference cost, we use the pricing figures in Table 7.

Table 7: Inference pricing used for cost analysis.

|  | Prompt Tokens (\$ / 1000 tokens) | Completion Tokens (\$ / 1000 tokens) |
|---|---|---|
| `gpt-3.5-turbo` | 0.0005 | 0.0015 |
| `gpt-4o` | 0.0025 | 0.01 |
| `llama-3.1-8b-instruct` | 0.00005 | 0.00008 |

---

[5] openai.com/api/pricing
[6] groq.com/pricing
[7] https://vast.ai/pricing/gpu/A40

```
┌─ Game-of-24 Value Estimation Prompt ──────────────────────┐
│                                                            │
│  Evaluate if the given numbers can reach 24 (sure/likely/impossible) Follow the │
│  example format exactly. Only evaluate the last example.   │
│  {few shot examples}                                       │
│  ────────────────────────────────────────────────────────  │
│  {input}                                                   │
│                                                            │
└────────────────────────────────────────────────────────────┘
```

Figure 16: Value estimation prompt for `Game-of-24`. This prompt was only used to prompt base models.

## H    Model Fine-tuning and Serving

Table 8: Hyperparameters during STL training.

|      | warmup-steps | learning-rate | weight-decay | per-device-batch size | lora-r | lora-alpha |
|------|--------------|---------------|--------------|------------------------|--------|------------|
| **STL** | 10 | $2e^{-4}$ | 0.01 | 8 | 16 | 16 |

Table 9: Effect of $\gamma$ on Performance.

| $\gamma$ | 0.75 | 0.80 | 0.85 | 0.90 | 0.95 | 0.99 | 1.0 |
|----------|------|------|------|------|------|------|-----|
| **WebShop Success Rate** | 32.0 | 24.0 | 24.0 | 24.0 | 28.0 | 42.0 | 46.0 |

Fine-tuning the value model for STL is carried out on a single NVIDIA A40 GPU. We use LoRA finetuning [25] and use models provided by unsloth[8]. The hyperparameters used are in Table 8. We fine-tuned `Game-of-24` value models for 10 epochs and `WebShop` value models for 20 epochs due to the differences in difficulty for models to learn the format of the action and state representations. We find that we require this large number of epochs to learn both the rationale structure and a good representation of transition dynamics. We note that it is clear that overfitting is now happening since unseen task performance does improve across tasks.

Additionally, we serve base and fine-tuned models using vLLM [9] [33] for efficient value estimation of new states during search. We use `temperature` = 1.0 and `max_tokens` = 3192.

As mentioned in §3, we use $\gamma = 1.0$ based on prior work. We also ran a small hyperparameter tuning validation for $\gamma \in \{0.95, 1.0\}$ on a 50-example held-out validation set and found $\gamma = 1.0$ had an average reward of 73.5 compared to 72.7 for $\gamma = 0.95$.

Finally, we perform a small analysis on the effect of the discount factor $\gamma$ on search performance on the test. The results of the analysis with a `gpt-3.5-turbo` policy on `WebShop` are presented in Table 9. The affect of $\gamma$ on the text set is more stark.

## I    Significance Testing

In §4.1 and §4.2, we use the paired bootstrap test to test the statistical significance of our experimental results. Following [3], we set $b = 10^6$. For `WebShop`, we run the significance test twice: once for score (average reward) and a separate time for success rate.

