# OpenReview forum: "Language Models can Self-Improve at State-Value Estimation for Better Search"
_NeurIPS.cc/2025/Conference — NeurIPS 2025 spotlight_

### Official Review · Reviewer_C12m · 2025-07-01

**Clarity:** 2
**Significance:** 3
**Originality:** 2
**Rating:** 3
**Confidence:** 4

**Summary:**

This paper proposes self-taught lookahead (STL), a self-supervised method that leverages state-transition dynamics to improve a value model capable of effectively guiding language model-controlled search without requiring ground-truth rewards or human demonstrations. The method fine-tunes a value LLM using lookahead state transitions and generated rationales, enabling lightweight yet effective search guidance across multiple tasks.

**Questions:**

1.The paper does not clearly differentiate STL from existing Rewardn and Demo Free models. It seems the main change is adding rationales to trajectories as input for value prediction.

2.Consider adding a brief mention of the method’s limitations and broader impacts in the Conclusion.

3.While most equations are clear, some notations (e.g., Equation 6) could benefit from further clarification.

**Ethical Concerns:**

["NO or VERY MINOR ethics concerns only"]

**Final Justification:**

I have carefully reviewed the authors' responses and examined all the reviewers' questions and discussions. The authors have fully addressed all my concerns and provided additional experiments. Therefore, I will update my score accordingly.

**Limitations:**

The authors discussed some limitations and broader impacts.

**Quality:**

2

**Strengths And Weaknesses:**

Strengths:

1.Addresses the high cost of human labels in multi-step reasoning tasks, an important practical limitation in deploying LLM-based agents.

2.Evaluates STL across diverse tasks with clear baselines and ablation studies illustrating each component's contribution

Weaknesses:

1.The paper states that it uses a single lookahead step within tree search to generate self-improvement data but does not introduce how tree search itself is performed.

2.Rationales are a key part of STL, but their structure and role remain abstract in the paper.

3.While comparisons with Reflexion, LATS, and AgentQ are present, the paper could be strengthened by explicitly discussing comparisons with recent methods, especially in 2024, highlighting differences in effectiveness and generalization.

---

> ### Author Rebuttal · Authors · 2025-07-30
>
> Thank you for your detailed feedback! We are glad you found that our work addresses an “important practical limitation in deploying LLM-based agents”. Please find responses to your comments below:
>
> > The paper states that it uses a single lookahead step within tree search to generate self-improvement data, but does not introduce how tree search itself is performed.
>
> As detailed in Section 3.1 (lines 97-99), we use classical tree search algorithms (BFS and MCTS) during the data generation phase. These methods have been widely used in prior LLM search work [1, 2] and are detailed in Section 2. For algorithms like MCTS, which require a reward to guide search, we use an LLM-generated value as a proxy reward, as we detail in Section 4.1 (lines 141 - 143). As you have mentioned, when a node is visited during tree search, we perform a step of lookahead to help inform a better estimate of the state’s value, as shown in Figure 2 and Section 3.1.
>
> > Rationales are a key part of STL, but their structure and role remain abstract in the paper.
>
> The rationales learned during self-improvement take the following form: “{action} {outcome state} {natural language value rationale}” as mentioned in Section 3.1 (line 108) and depicted in the top right of Figure 2. Besides this lookahead structure, we do not impose any format for the natural language value rationale, but rather bootstrap a base instruction-tuned model. However, we agree it would be helpful to see examples of these rationales. We provide an example of the training data below, which we will add to the Appendix:
>
> ##### **Task and Current Trajectory:**
>
> Instruction: i'm looking for a pair of women's workout shorts with a drawstring waist. i need them to be extra large and in light gray, and price lower than 60.00 dollars
> [Search]
>
> Action: search[women's workout shorts drawstring waist extra large light gray < 60 dollars]
>
> Observation: *removed for brevity*
>
> Current Action: click[B09T3PJM1R]
>
> Current Observation: *removed for brevity*
>
> ##### **STL Output:**
>
> I will evaluate the best successor state from the current state:
>
> Best Next Action: click[x02c-gray]
>
> Observation of Best Successor State: You have clicked x02c-gray.
>
> Reflection of the Best Successor State: The last action selects the color 'x02c-gray' for the item B09T3PJM1R.
> Based on the observation, this product's color is indeed gray, which matches the specified criteria.
> Therefore, this product matches one of the attributes mentioned in the task.
> The last action and observation thus capture a step that selects an attribute mentioned in the instruction, but not all attributes mentioned (specifically the size attribute) are currently selected.
> Thus, the correctness score is 6.00 / 10.00.
>
>
>
> > While comparisons with Reflexion, LATS, and AgentQ are present, the paper could be strengthened by explicitly discussing comparisons with recent methods, especially in 2024.
>
> We agree that comparison with recent methods is crucial to contextualize our work. However, we note that methods like LATS and AgentQ are from 2024, with the former appearing at ICML 2024 and the latter appearing on arXiv in August 2024. We also compare to search with a reasoning value model r1-distill-llama-8b, which was released in January 2025. In the HotpotQA domain, we also compare to R1-Searcher (released in March 2025). However, we do agree that it would be interesting to include additional comparisons with recent models. Therefore, we ran an experiment using a gpt-3.5-turbo policy with a qwen3-8b reasoning value model that was released in late April 2025. The results on the 50-example WebShop test set are provided in the Table below:
>
> | **Model**                               | **Average Reward**  | **Success Rate** | **Cost** |
> |-----------------------------------------|--------------------|------------------|----------|
> | qwen3-8b (thinking mode enabled)        | 68.6   | 34.0            | $3.80     |
>
> We will add these results to the camera-ready version of the paper.
>
>
> > The paper does not clearly differentiate STL from existing Reward and Demo Free models. It seems the main change is adding rationales to trajectories as input for value prediction.
>
> As depicted in Figure 1 and explained in the Introduction (lines 27 - 29), other Reward and Demo Free methods are *inference-only methods* that cannot *improve* performance through learning and are therefore constrained to the performance of the off-the-shelf LLM used for search. In contrast, STL enables improvement even in the data-scarce Reward and Demo Free setting by explicitly learning in the form of supervised-finetuning on rolled-out lookahead and generated rationales. As we demonstrate across tasks, this value model improvement yields better value prediction and superior downstream search performance.
>
> > Consider adding a brief mention of the method’s limitations and broader impacts in the Conclusion.
>
> Thank you for the suggestion! We note that we have discussed Limitations in Appendix C, but will add some of this discussion to the conclusion in the camera-ready version of the paper.
>
>
> > While most equations are clear, some notations (e.g., Equation 6) could benefit from further clarification.
>
> Thank you for the suggestion; we will add references to Figure 2 when discussing Equation 6 in Section 3.3 to help clarify the notation. We will also reiterate the concatenation notation, which was first introduced in Section 3.1, again in Section 3.3.
>
>
> [1] Andy Zhou et al. 2024. “Language Agent Tree Search Unifies Reasoning, Acting, and Planning in Language Models.” Proceedings of the 41st International Conference on Machine Learning
>
> [2] Shunyu Yao et al. 2023. “Tree of Thoughts: Deliberate Problem Solving with Large Language Models”. Advances in Neural Information Processing Systems 36 (NeurIPS 2023) Main Conference Track

---

> ### Author Response · Authors · 2025-08-05
>
> Thank you again for your thoughtful review. As we approach the end of the author-reviewer discussion period, we would like to confirm whether we have addressed all of your concerns in our rebuttal. Please let us know if there is anything else we can clarify.

---

### Official Review · Reviewer_XxXy · 2025-07-02

**Clarity:** 3
**Significance:** 3
**Originality:** 4
**Rating:** 5
**Confidence:** 4

**Summary:**

This paper introduces a method called “self-taught lookahead” (STL) which can be thought of as a form of value iteration applied to language models. In STL, the model is prompted to estimate state values along with rationales for multiple successor states, and then actions with the highest value are used to form a finetuning dataset consisting of (state, action, successor state, rationale, value). The model is finetuned, and at inference time greedy search is used to select subsequent actions. Experiments are conducted on three benchmarks (WebShop, HotpotQA, and Game of 24) using a finetuned llama-3.1-8b-instruct as the value model, demonstrating STL to work as well as or better than gpt-4o as the value model.

**Questions:**

### Clarifications

* I didn’t quite understand how the generation of rationales and values works. In Section 3.1 and Algorithm 1, it sounds as though: (1) values are first generated on their own for several actions, leveraging ground-truth transitions, and then (2) rationales are generated given the actions and outcomes. But then there is only a single prompt (e.g. Figure 9\) for value estimation, which first estimates the rationale, and then the value. So I’m confused about what the actual order of estimation is.
* I don’t think I fully understand what the algorithm looks like at inference time. It would be helpful to include an additional algorithm clarifying this. Is it that several actions are drawn from the base model, then these are each evaluated by the value model, and the best one is picked?
* It’s not clear what the dataset sizes were for finetuning. It would be helpful to include this detail in Appendix H, along with maybe some other statistics about the dataset generation (e.g. how many tasks, how many rollouts from each task, how long those rollouts were, what was the success rate/score of the training data, etc.). Additionally, please report other hyperparameters like context size, maximum generation length, temperature, etc.

### Methodological questions

* I didn’t understand why the paper did not include baselines for just vanilla, out-of-the box LLMs with no search at all. The paper includes baselines with out-of-the-box models that are used in the same greedy search fashion as STL, but there does not seem to be anything which just greedily selects actions with no value estimation. I think this is also an important comparison to include with respect to the efficiency argument, since such a model will have a much smaller inference time cost. It would be great if you could add such model-free baselines with GPT-3.5-Turbo and GPT-4o for comparison.
* “We train value models at each position (depth) in the trajectory” (line 594\) → this was a really surprising methodological choice to me, and seems really strange. Couldn’t you just condition the model on what depth it’s at? Also, the fact that you actually have 4x finetuned models is a big overhead (having to serve 5 models in total\!) and needs to be mentioned in the main text when discussing computational efficiency.
* I found the choice to not use ground-truth rewards at all to be curious (especially given that you *do* use the ground-truth transition function). Wouldn’t you want to at least leverage these for final states? Given that you limit trajectories to be of max 5 on WebShop and then force a buy action, it seems like you should be able to include the rewards at these terminal states. The lack of grounding in real rewards might explain why multiple iterations of STL don’t seem to help on WebShop.
* “We fine-tuned Game-of-24 value models for 10 epochs and WebShop value models for 20 epochs” (line 681\) → this sounds like a *lot* of epochs and I’m fairly worried about overfitting. Did you do any analysis to verify that overfitting is not happening?
* Why did you use MCTS to generate the training data? Does it work better than BFS or just Monte Carlo rollouts? It would help to motivate this choice more, given that it is a more complicated algorithm.

### Other things

* While it’s great that the paper includes tests for statistical significance, it would also be good to include error bars / confidence intervals.
* I think there should really be some more discussion of RL-trained reasoning agents in the Related Work section, i.e. R1, o1/o3, Gemini 2.5, Claude 3.7/4, etc., given that this approach is the current zeitgeist.
* What’s the budget you use for MCTS?

**Ethical Concerns:**

["NO or VERY MINOR ethics concerns only"]

**Final Justification:**

In my original review, I had some clarification questions and some concerns over some of the methodology. I think the authors did a good job addressing these in their rebuttal, especially with the inclusion of the ReAct baseline. Even after the rebuttal I do still think the task-specific prompts limit the significance of the work, but as the authors pointed out this is common in prior work too so I don't think it should be counted against the paper for acceptance.

**Limitations:**

yes

**Quality:**

3

**Strengths And Weaknesses:**

Strengths

* Significance: Training LLMs to work well in multistep, agentic settings is an important and active area of research. The demonstrated results are impressive, showing that self-improvement can enable smaller models to work as well as larger, more expensive models.
* Clarity: the paper is well-written with the results clearly presented. There are a number of insightful points in the paper that I think will be appreciated by others working in the space of reasoning/planning.
* Quality: the paper presents an impressive range of experiments, including a wide range of appropriately chosen baselines, informative ablations, and extensive analyses on efficiency and scaling. Statistical significance tests are conducted.
* Originality: the proposed method is a clever application of older ideas in RL to the LLMs, and in particular leverages some of the strengths of LLMs in ways that were impossible to do in the classic setting (i.e., including rationales for the value estimates).

Weaknesses:

* Clarity: there were a few points of confusion I had about the methods. While the appendix is already great, I think it could be further improved with a bit more detail. I discuss my particular points of confusion in more detail in the “Questions” section.
* Quality: there are a few methodological choices that seemed a bit strange to me, or which perhaps I didn’t fully understand. I discuss these further in the “Questions” section below.
* Significance: the method requires task-specific prompts for value estimation, which limits its generality.

---

> ### Author Rebuttal · Authors · 2025-07-30
>
> Thank you for your thorough review! We are glad that you found STL to be a “clever application of older ideas in RL to the LLMs” with “impressive results” over an “impressive range of experiments”. We have addressed your comments below:
>
> > How does the generation of rationales and values work?
>
> As you mentioned, using the base model, values are first generated for actions encountered through a step of lookahead (performed with the ground truth transition function) while visiting nodes during tree search. These values are generated using the prompt in Figure 9. The result of this lookahead is then used to construct formatted training examples that textually capture the lookahead process, as shown in the top right of Figure 2. After training on this dataset, we obtain an STL value model. When we perform inference with this model, we *do not use* the prompt in Figure 9, but instead use a prompt without demonstrations that looks like the training examples in the top right of Figure 2, but with the relevant information about the state being evaluated. The STL value model then simulates a step of lookahead, as shown in the bottom right of Figure 2 and mentioned in Section 3.3. We will add more discussion about the distinction between these two prompts in Section 3.3 and the Appendix.
>
> > I don’t think I fully understand what the algorithm looks like at inference time. Is it that several actions are drawn from the base model, then these are each evaluated by the value model, and the best one is picked?
>
> As discussed in the previous response, during inference time, a step of lookahead is simulated by the STL value model to generate value judgments. Yes, in greedy search, the policy generates multiple possible actions from the current state. The STL value model then evaluates each of these actions, and the action with the highest estimated value is selected. We will make this clear in the revised version when it is discussed in Section 4.1, and also add a full algorithm for inference like the one we have included for training in the Appendix.
>
> > It’s not clear what the dataset sizes were for finetuning. It would be helpful to include this detail in Appendix H, along with maybe some other statistics about the dataset generation
>
> The size of the WebShop training set was 1161, collected from rolling out search trees for 50 tasks in the training set. The size of the HotpotQA training set was 2708, collected from rolling out search trees for 500 tasks in the training set. As mentioned in Section 4.2, we require more tasks and training examples for HotpotQA since it has lower action diversity. We used a maximum generation length of 3192 tokens (rationales for most models were only a few sentences, but we raised this token limit to account for reasoning models) and a temperature of 1.0. We will add these to the Appendix of the camera-ready version of the paper.
>
> > I didn’t understand why the paper did not include baselines for just vanilla, out-of-the-box LLMs with no search at all. It would be great if you could add such model-free baselines with GPT-3.5-Turbo and GPT-4o for comparison.
>
> Thank you for the suggestion! We have computed the performance of ReAct prompting (without search) with these models. Their results and costs on the 50-task WebShop test set are provided below:
>
> |**Model**|**Average Reward**|**Success Rate**|**Cost**|
> |---|---|---|---|
> |gpt-3.5-turbo|68.9|36.0|$0.18|
> |gpt-4o|70.0|36.0|$1.09|
>
> These figures support the result that there does not seem to be a significant difference between using a gpt-3.5-turbo and gpt-4o policy model (Section 4.1). We will add these new results to the camera-ready version of the paper.
>
>
> > “We train value models at each position (depth) in the trajectory” (line 594). Couldn’t you just condition the model on what depth it’s at?
>
> Yes, in preliminary experiments, we did condition on the current trajectory depth, but found that the fine-tuned llama-3.1-8b-instruct models were unable to properly differentiate between similar actions taken at different depths, which led to poor simulated lookahead and poor search performance. Therefore, we decided to adopt the current approach. We do agree that conditional modeling may work, but it likely requires training a larger LLM. However, we did not have the computing resources to fine-tune the larger llama-3.1-70b-instruct model. Additionally, while hosting multiple models during inference does increase computational costs, the total compute needed is *still far less than that required to host the models from other similar papers*. For instance, AgentQ produces a 72b policy model, which has more than double the number of parameters as our setup. We will add this discussion to Section 5 of the paper.
>
> > I found the choice not to use ground-truth rewards at all to be curious (especially given that you do use the ground-truth transition function)
>
> As we discussed in the introduction (lines 23 - 25), ground truth reward (such as web task completion) is expensive and time-consuming to collect, as it often has to be manually annotated by humans. However, the ground truth transition function is almost always provided directly by the environment, as we note in the introduction (lines 39 - 41), which makes its use during training and inference a realistic and common assumption. For instance, in web-based tasks, the browser itself determines how transitions occur. STL is designed to address the common setting where practitioners desire to improve LLM search performance on a new task, but are unable to collect ground truth rewards. While the benchmarks we use for evaluation have ground truth rewards, in order to simulate this data-scarce setting, STL does not use the provided reward.
>
> > Did you do any analysis to verify that overfitting is not happening?
>
> On WebShop and HotpotQA, we found that more epochs were needed for models to learn the lookahead format of rationales. Since STL performance showed significant improvements over using the original llama-3.1-8b-instruct model as the value model on *unseen tasks*, we can be confident that significant overfitting is not occurring on these tasks.
>
> However, on the Game-of-24 task, we do see a decrease in performance after the first step of self-improvement. Through an experiment in the response to TNro, we found that this decrease was likely caused by too few tasks seen during each step of self-improvement. However, we also tried training with only a single epoch instead of 10 and found that *performance slightly improved from 48% to 52%* after 50 tasks were seen during self-improvement. Thank you for pointing this out; we will add this result to the camera-ready version of the paper.
>
> > Why did you use MCTS to generate the training data? Does it work better than BFS or just Monte Carlo rollouts?
>
> STL training data can be generated with any tree search algorithm since data generation only requires a step of lookahead to be performed after visiting each node. As we note in Section 4.1, we used MCTS rollouts on WebShop since prior work has shown empirical advantages to using MCTS as opposed to other algorithms. On Game-of-24, we use BFS to perform rollouts as was canonically done in the Tree of Thoughts paper [1]. Due to experimental costs, we were not able to conduct an ablation on the tree search algorithm used, but the results of such a study would likely be dependent on the task.
>
>
> > While it’s great that the paper includes tests for statistical significance, it would also be good to include error bars
>
> We are glad that you found our use of statistical significance tests helpful. Below, we have computed error bars on the full WebShop test set, which we will add to the Appendix of the camera-ready version:
>
> | **Policy** | **Value Model**| **Average Reward** | **Success Rate** |
> |-|-|-|-|
> | gpt-3.5-turbo  | llama-3.1-8b-instruct| 67.7 ± 2.26 | 26.4 ± 3.86|
> | | deepseek-r1-distill-llama-8b    | 66.3 ± 2.30| 24.6 ± 3.78|
> |  | gpt-3.5-turbo | 70.6 ± 2.43 | 35.6 ± 4.23|
> |  | gpt-4o | 71.5 ± 2.51| 40.6 ± 4.33|
> | gpt-4o| llama-3.1-8b-instruct  | 67.2 ± 2.30| 25.8 ± 3.84|
> || deepseek-r1-distill-llama-8b| 66.5 ± 2.33| 25.6 ± 3.83|
> || gpt-3.5-turbo| 72.4 ± 2.40| 38.8 ± 4.29|
> | | gpt-4o| 71.4 ± 2.49| 40.8 ± 4.35 |
> | **––– STL Results –––** |
> | gpt-3.5-turbo  | llama-3.1-8b-instruct-STL | 72.8 ± 2.32 | 36.6 ± 4.22|
> | gpt-4o | llama-3.1-8b-instruct-STL | 74.2 ± 2.38 | 40.6 ± 4.30|
>
> > I think there should be some more discussion of RL-trained reasoning agents in the Related Work section
>
> Thank you for the suggestion! We will add a discussion in the related work section about recent work [2] that shows that models explicitly trained for agentic use, like Deep Research, greatly outperform general-purpose RL-trained reasoning models like o1. This work corroborates findings in our paper that search with specialized value models improved with STL outperforms RL-trained reasoning models like r1-distill and qwen-3.
>
> > What’s the budget you use for MCTS?
>
> We run MCTS for 30 iterations, which is the same setting used by the LATS baseline. We will clarify this in Section 4.
>
> > The method requires task-specific prompts for value estimation, which limits its generality.
>
> Generality is an important consideration. However, almost all prior LLM agent work uses task-specific prompts for action generation and/or value estimation, including Reflexion, ReAct, LATS, and ToT.
>
>
> [1] Shunyu Yao et al. 2023. “Tree of Thoughts: Deliberate Problem Solving with Large Language Models”. Advances in Neural Information Processing Systems 36 (NeurIPS 2023) Main Conference Track
>
> [2] Jason Wei et al. 2025. BrowseComp: A Simple Yet Challenging Benchmark for Browsing Agents. OpenAI.

---

> > ### Comment · Reviewer_XxXy · 2025-08-05
> > **Response to authors**
> >
> > Thanks very much for all the clarifications, they are very helpful. It will definitely improve the paper to include these details in the appendix for other readers too. Overall my concerns have been fully addressed and I will keep my score as a "clear accept" and raise my confidence to a 4.
> >
> > ### ReAct baselines
> >
> > Thanks a lot for running these baselines, the comparisons are very interesting—I think it makes the result even stronger that the choice of value function is super important (i.e. that poor value functions will barely improve performance over no search at all)!
> >
> > ### Ground truth rewards
> >
> > Thanks for the explanation. I think it might be worth expanding on this slightly in the main text—e.g. you could give an example of where the ground-truth reward is hard to annotate but where the transition function is easy to simulate. I realize you do say that the rewards are hard to get ("collecting ground-truth rewards or human demonstrations may not be possible in every environment, and can oftentimes be costly") but it would be more convincing to motivate *why* that's the case.

---

> > > ### Author Response · Authors · 2025-08-05
> > >
> > > Thank you for your thoughtful feedback and for raising your confidence to 4. We are glad the rebuttal has fully addressed your concerns. We agree that the new ReAct baselines reinforce our motivation, and we will highlight them in the camera-ready version. We also appreciate your suggestion to illustrate why collecting ground-truth rewards is difficult and why transitions are usually easy to simulate. In the introduction, we plan to add a more explicit web task example, which we agree will strengthen the paper’s motivation and clarity.

---

### Official Review · Reviewer_rcW5 · 2025-07-03

**Clarity:** 4
**Significance:** 4
**Originality:** 4
**Rating:** 5
**Confidence:** 5

**Summary:**

In this work, the authors propose Self-Taught Lookahead (STL), a self-supervised method that leverages state transition dynamics to improve a value model without using any labeled data. The proposed method is used to train the Llama-3.1-8b-instruct model as a value model, which can then be used with any search algorithm at inference to improve the performance of a Language Model-based Agent. Authors conduct extensive experimentation using different value models (trained and training-free) to demonstrate the efficacy of the learned value model. The experimental analysis is conducted on three tasks: WebShop, Multiple-Choice QA (HotPotQA), and Math Puzzles (Game of 24). Experiments show that a 8B parameter learned value model can match gpt-4o performance in all the tasks. The proposed training approach is comparatively lightweight, resulting in significant experimental efficiency.

The paper is well structured and clearly written. Numerous approaches have been proposed to utilize and train LLMs as value models. To the best of my knowledge, this work presents a novel approach that demonstrates efficiency and performance gains over contemporary methods. The experimental analysis is exceptionally well done, with strong and relevant baselines. The authors provide statistically significant results, which is rare in this research space and is appreciated.

Overall, this is a well thought out and well executed piece of work, and thus **I vote for a clear accept**.

**Questions:**

## Questions

- How does the branching factor apply to Greedy search? Is the next action sampled $n$ times, and the best value is selected?
- Were there any experiments conducted with different search algorithms using the trained value model?
- In Figure 4, what is the difference between open-source and closed-source tokens?
- How does the size of the action space affect the performance of the proposed method? Any experiments on this topic would significantly strengthen this paper.
- One notable result from Table 1 is that the STL with Greedy search performance is significantly lower compared to the AgentQ (trained policy model). It could be interesting to explore the interaction between these trained policy models (even though they are not reward-free) and the trained value model. Additionally, I am uncertain about the contribution of the GPT-4v value model used in the AgentQ method. Any discussion on this would be greatly appreciated.

**Ethical Concerns:**

["NO or VERY MINOR ethics concerns only"]

**Final Justification:**

Authors have addressed all of my concerns during the rebuttal phase. My main concern was limited frontier model coverage; however, due to budget constraints, authors are not able to run experiments using these models. In my view, this is a minor issue that can be fixed in later iterations or with community involvement. The core idea explored in this work remains useful, and thus, I vote for accepting this paper with a rating of 5.

**Limitations:**

Yes.

**Paper Formatting Concerns:**

No.

**Quality:**

4

**Strengths And Weaknesses:**

## Strengths

- The problem addressed in this work is well-positioned within the current research landscape.
- The proposed method is comparatively lightweight and does not require human-labeled data, but still outshines the strong baselines.
- The experimental analysis demonstrates the statistically significant performance gains while being efficient in terms of cost and search steps.
- The paper is well-written and easy to follow.

## Weaknesses

- One potential weakness of this work is the optional requirement of task filters while creating the training data. While it is optional, I assume that for complex agent environments, this may prove to be challenging to build.
- For the Reward and Demo Free setting, authors used slightly outdated models (except R1-distilled) as value models. I assume this is done to maintain budget constraints and to provide a fair comparison with existing baselines. However, it will be interesting to see the comparison between newer models, such as the Claude-3.5/3.7/4-Sonnet or Gemini-2.5-Pro, and the trained value model.
While the datasets selected for this work are suitable, it would have been beneficial to see a more open-ended setting with a relatively larger and more fluid action space, such as software engineering environments.

Overall, the strengths far outweigh the weaknesses, and all the weaknesses can be addressed in future work.

---

> ### Author Rebuttal · Authors · 2025-07-30
>
> Thank you for your thorough review! We are glad that you find that our work is “well thought out and well executed” and “vote for a clear accept”. We have addressed your comments below:
>
> > One potential weakness of this work is the optional requirement of task filters while creating the training data. While it is optional, I assume that for complex agent environments, this may prove to be challenging to build.
>
> The task filters we used in this paper were simple string checks to ensure that the training data did not contain malformed rationales, as we indicate in Appendix D.3. For instance, we verify that the training data rationales follow the correct format such as including the phrase “Thus, the correctness score is” before providing a score estimate so that the value can be easily and accurately parsed. We do not foresee many cases where an especially intricate task filter is needed, but in these cases, an LLM could be used to filter data as has been proposed by prior work, including [1].
>
> > For the Reward and Demo Free setting, authors used slightly outdated models (except R1-distilled) as value models… However, it will be interesting to see the comparison between newer models, such as the Claude-3.5/3.7/4-Sonnet or Gemini-2.5-Pro, and the trained value model.
>
> Thank you for the suggestion. Unfortunately, we do not have the budget to run these newer proprietary models at the moment. However, we agree it would be interesting to see results with newer models, so we ran an experiment using a gpt-3.5-turbo policy with a qwen3-8b reasoning value model that was released in late April 2025. The results on the 50-example WebShop test set are provided in the table below:
>
> | **Model**                               | **Average Reward** | **Success Rate** | **Cost** |
> |-----------------------------------------|--------------------|------------------|----------|
> | qwen3-8b (thinking mode enabled)        | 68.6   | 34.0            | $3.80     |
>
> These figures are quite similar to performance with a r1-distill-llama-8b value model. This result may indicate that while RL-trained reasoning models have steadily improved performance on math reasoning tasks over the last few months, they still struggle to perform calibrated state-value estimation out-of-the-box. We will add this result to the camera-ready version of the paper.
>
> > How does the branching factor apply to Greedy search? Is the next action sampled $n$ times, and the best value is selected?
>
> Yes, during greedy search, the branching factor dictates how many actions are sampled from the policy. The value model then evaluates each action and selects the one with the highest value. We will add a couple of sentences in the revised version to make this clearer when it is discussed in Section 4.1, and also add a full algorithm for greedy inference in the Appendix.
> > Were there any experiments conducted with different search algorithms using the trained value model?
>
> We found that MCTS with the base llama-3.1-8b-instruct value model was quite resource-intensive in terms of GPU hours required. Since STL with greedy search already outperformed MCTS with the base instruct model, we decided against running more expensive search algorithms like MCTS with the STL value model as well, but we believe this may be interesting to explore in future work.
>
> > In Figure 4, what is the difference between open-source and closed-source tokens?
>
> Open-source tokens are prompt and completion tokens from open-source models like llama-3.1-8b-instruct, r1-distill-llama-8b, etc. Closed-source tokens are prompt and completion tokens used with closed-source proprietary models like gpt-3.5-turbo, gpt-4o, etc. We agree that this distinction could use some clarification; we will update the wording in the Figure 4 caption and in Section 5.1 to make this difference clear.
>
> > How does the size of the action space affect the performance of the proposed method? Any experiments on this topic would significantly strengthen this paper.
>
> As discussed in Section 4 and Appendix D.2, achieving strong performance with STL requires high action diversity (a large number of possible actions at each step, or a large action space). When action diversity is low, the effect of the value model on search performance is diminished, and it is necessary to roll out more tasks to obtain enough data to fine-tune the value model. For example, Section 4.2 (lines 202 - 204) notes that due to its low action diversity, HotpotQA shows smaller relative gains from STL compared to WebShop and also requires rollouts of 500 tasks (10 times more than WebShop) to get a large enough dataset for fine-tuning. Unfortunately, it is difficult to design an experiment to measure the effect of the size of the action space on performance since different domains intrinsically have different action space sizes. One could artificially reduce the action space size by restricting the set of possible actions, but doing so would introduce confounders since search with an off-the-shelf value model would also likely decrease in performance with such a restriction. Therefore, we believe that comparing gains across domains, such as between WebShop and HotpotQA, is the best way to understand how the size of the action space affects STL performance. As mentioned in the response to TNro, we will add some discussion about this in Section 3 and the Appendix.
>
> > One notable result from Table 1 is that the STL with Greedy search performance is significantly lower compared to the AgentQ (trained policy model). It could be interesting to explore the interaction between these trained policy models (even though they are not reward-free) and the trained value model. Additionally, I am uncertain about the contribution of the GPT-4v value model used in the AgentQ method.
>
> This is an interesting suggestion! To our knowledge, the 72B AgentQ model has not been released, so we unfortunately cannot run this experiment at this time. However, if this model is released by the authors, we will run this interaction experiment and add the results to the camera-ready version of the paper. The GPT-4v value model that AgentQ utilizes is used to provide feedback on trajectories to guide MCTS search. We will clarify this when we discuss AgentQ in Section 4.1 of the paper.
>
>
> [1] Pei Ke et al. 2024. “CritiqueLLM: Towards an Informative Critique Generation Model for Evaluation of Large Language Model Generation.” Proceedings of the 62nd Annual Meeting of the Association for Computational Linguistics.

---

> > ### Comment · Reviewer_rcW5 · 2025-08-01
> > **Response to Authors**
> >
> > Thank you for the detailed response. The author's response has addressed all of my questions and concerns about the work. Since I have already voted for clear acceptance, I will keep the score and increase my confidence to 5.
> >
> > I have some comments on the effect of the size of the action space. I agree with the authors that designing a controlled experiment for this setting is difficult in practice. However, what I meant by this comment in the original review was how STL works for very large action space environments. Although the WebShop environment has a kind of hierarchical action space, i.e., there are only nine broader actions that you can take (click, choose, search, etc.), each of these actions has further options, the set of core actions is still limited.
> > It would be interesting to explore how STL generalizes and performs in environments with action spaces of over 100 actions [1]. I assume that since STL requires sampling actions with high diversity, as the size of action spaces increases, so will the computational requirements.
> > Please note that, in my view, these experiments will strengthen the claims made in the paper but are not required for acceptance of the work.
> >
> > [1] Landers, M., Killian, T.W., Barnes, H., Hartvigsen, T., & Doryab, A. (2024). BraVE: Offline Reinforcement Learning for Discrete Combinatorial Action Spaces.

---

> > > ### Author Response · Authors · 2025-08-02
> > >
> > > Thank you for your response - we are glad the rebuttal fully addressed your concerns and appreciate your strong endorsement of the paper’s acceptance!
> > >
> > >
> > > Regarding your clarification on large action spaces: we agree that while WebShop has a large number of actions, they’re hierarchically structured, and their linguistic form enables LLMs to draw on their strong priors. This grounding seems to be key to STL’s success and could extend to other tasks with large action spaces, provided actions are semantically aligned with their effect.
> > >
> > >
> > > In contrast, environments like the high-dimensional CoNE environment that BraVE uses for evaluation pose different challenges with flat, ungrounded, combinatorial action spaces. Without meaningful linguistic structure, STL would likely require more sampling and computational resources, as you mentioned.
> > >
> > > Exploring how to bridge this gap in traditional RL environments, perhaps by learning intermediate grounded representations or using learned action names, seems like an exciting direction for future work. Thank you again for raising this, as it is an important consideration and one we hope the community continues to explore. We will include some discussion about this in the camera-ready version of the paper.

---

> > > > ### Comment · Reviewer_rcW5 · 2025-08-04
> > > > **Response to Authors**
> > > >
> > > > Thank you for addressing all the concerns. I look forward to reading the discussion on how the proposed method can be used in environments with limited linguistic features in the action space. I will keep my rating and increase my confidence.

---

### Official Review · Reviewer_TNro · 2025-07-06

**Clarity:** 3
**Significance:** 3
**Originality:** 3
**Rating:** 5
**Confidence:** 4

**Summary:**

This paper offers an algorithm to improve LLM-based value function estimates for multi-step reasoning tasks via “self-taught lookahead”: exploring state transitions via tree search, estimating the values at the new states, and performing a Bellman backup to obtain a better estimate of the value of the current states. Iterative finetuning via this approach trains a better LLM value function, which then is paired with LLM-simulated lookahead for greedy value-based action selection at inference time. The paper presents empirical results across Webshop, HotPotQA, and Game-of-24 that indicate improve task performance due to the fine-tuned value function.

**Questions:**

Could you please provide some context to help me understand the experimental results on Webshop--the baseline numbers for Reflexion are far worse than reported in prior work. See my prior comments for discussion of both ReAct and Reflexion numbers--which should be a lower bound for search-based success rates. This is the place where clarification could most help change my evaluation score.

I would appreciate a longer discussion of Figure 3—is there an experiment you can use with a stronger value function for lookahead states that prevents the earlier decline in performance on unseen tasks?

Practitioners would highly benefit from guidelines for situations in which this approach is applicable vs where it is not--could you please include a discussion of the types of reasoning tasks where LLMs provide good value estimates at terminal states? (Which seems to be the key requirement). This feels like a critical discussion to include and would also help raise my score.

I would suggest including greater detail for some implementation details--number of iterations of Reflexion, source of Human performance baseline, etc.

**Ethical Concerns:**

["NO or VERY MINOR ethics concerns only"]

**Final Justification:**

Prior to rebuttal, my main concerns were with 1) the weak non-search baseline numbers, 2) the decline in task performance seen over iterations in Figure 3, and 3) some missing details on where the model is applicable, train-time costs, etc.

The rebuttal and discussion have addressed all three concerns, though I agree with reviewer rcW5's comment that the action spaces considered are hierarchically structured and do not offer the high dimensionality of "flat", large action spaces.

**Limitations:**

See note in weaknesses on the assumption that the value estimates from the LLM for terminal states is good—practitioners would highly benefit from guidelines for situations in which this approach is applicable vs where it is not.

**Quality:**

3

**Strengths And Weaknesses:**

**Strengths**:

The problem setting is valuable—LLM agents may be deployed in real-world settings where neither rewards nor human demonstrations are available, and in these settings environment transition dynamics provide a self-supervised learning signal.

Comparisons to other search-based methods are thorough, and indicate performance on par with/exceeding more powerful baseline LLMs.

Comparisons of test-time token usage and cost are informative.

**Weaknesses**:
The non-search baselines for Webshop are not strong—looking at Figure 5 in the ExpeL paper (https://arxiv.org/abs/2308.10144), ReAct with GPT3.5 turbo achieves 35% success rate on Webshop, and Reflexion with 1 and 3 rounds improves performance to 43 and 48% respectively—while this paper’s Reflexion baseline with the same LLM (please report number of Reflexion iterations) achieves 16.4% success rate. The performance of tree search-based methods should be an upper bound on the performance of Reflexion—so I would expect these success rates to all be higher than 48%.

A critical component of the problem setup is the assumption that at later/terminal states, the value function estimates provided by the LLM is “good”—so that the backup operation improves value estimates at earlier states. It seems important to include a discussion of the types of tasks where LLMs are good estimators of value at terminal states—to motivate situations in which this approach is applicable.

The discussion of the presence/absence of human demonstrations in Figure 1 feels extraneous given that, outside the IL baseline, all other discussion of problem setting is whether or not environmental rewards are present at train/test time.

Missing a discussion of train-time token usage when purely discussing test-time compute usage.

How did you obtain the Human Expert row in Table 1?

---

> ### Author Rebuttal · Authors · 2025-07-30
>
> Thank you for the detailed feedback! We are glad that you found “the problem setting valuable” and our analysis “comprehensive”. We have addressed your comments below:
>
> > The non-search (Reflexion) baselines for Webshop are not strong
>
> Thank you for pointing this out! We originally ran these experiments using the official Reflexion Github repository with three iterations, changing only the model from text-davinci-003 (a deprecated text completion model used in the original paper) to gpt-3.5-turbo. During the author rebuttal period, we found the following:
> * Other researchers had obtained similar Webshop performance to our reported numbers (~15%) using the unchanged official implementation with gpt-3.5-turbo (see Github issue #49).
> * Others also noted that changes to the prompts were needed to adapt the framework to conversational models to see improved performance in the 30-40% range (see Github issue #48).
>
> * Another researcher also identified a bug in the WebShop implementation that prevented the use of memory from prior iterations (see Github issue #36). In the discussion of this issue, the first author of the Reflexion paper acknowledged that this bug may have caused the lack of improvement of the Reflexion agent on WebShop that was reported in Appendix B.1 of the original Reflexion paper [1].
>
> After modifying the prompts and patching this memory bug, the ReAct success rate on the 50-task test set is *36%* and the Reflexion success rate is *46%* after three iterations, which is similar to the performance reported in the ExpeL paper. For Reflexion, we also recomputed the cost ($0.53) and number of expanded states (488), which are both similar to the current figures in the paper.
>
> We will update these results in the camera-ready version of the paper and state the changes made to the original repository to achieve these results in the Appendix. We note that this change *does not affect the conclusions of the paper* since Reflexion and STL belong to different reward settings.
>
>
> > The performance of tree search-based methods should be an upper bound on the performance of Reflexion—so I would expect these success rates to all be higher than 48%.
>
> The performance of LATS, a tree search-based method, is 38% as reported in the original LATS paper. After updating the Reflexion baseline, though their average reward scores at 77.2 and 75.9 are similar, LATS does seem to underperform Reflexion. We agree that LATS is more computationally expensive than Reflexion. However, due to differences in the mechanisms of these two methods, it is not necessarily the case that LATS performance is lower bounded by Reflexion performance. For instance, Reflexion enables the LLM policy to make wholesale changes to any steps in its trajectory after reflecting on previous failures. On the other hand, while LATS provides its policy and value LLMs with reflections on previous trajectories, it still relies on the traditional (non-neural) mechanics of MCTS. For example, LLM reflection *is not involved during node selection*, and instead, the traditional Upper Confidence bounds applied to Trees (UCT) heuristic is used. We will add this discussion to Section 4 of the revised version of the paper to help contextualize these results.
>
> We would also like to mention that, as noted in the introduction and throughout the paper, LATS and Reflexion are Reward-Guided Inference methods that have access to ground truth reward during inference. However, STL is a Reward and Demo Free method, which *does not have access to ground truth reward* during search. Therefore, since LATS and Reflexion involve a different level of supervision compared to STL, *they are not true baselines*, but rather help contextualize results of the Reward and Demo Free setting with the existing literature.
>
>
> > I would appreciate a longer discussion of Figure 3—is there an experiment you can use with a stronger value function for lookahead states that prevents the earlier decline in performance on unseen tasks?
>
> After some investigation, we believe that this early decline in unseen task performance is due to the limited number of tasks the value model sees during training during the first couple of iterations. Specifically, if value models are not exposed to enough actions and their lookahead values during training, they fail to generalize well to unseen tasks. Limiting the number of tasks per iteration also limits the quantity and diversity of actions and values seen. When we double the number of tasks the value model sees during each iteration from 25 to 50, we can avoid a decline in performance (accuracy increases in the first iteration from 38.0% to 48.0%), as shown in the table below:
>
> | **Num Seen Tasks During Self-Improvement** | **Performance: Tasks Seen Per Iteration=25** | **Performance: Tasks Seen Per Iteration=50** |
> |----------------------------------------|----------------------------------------------|----------------------------------------------|
> | 0                                      | 38.0                                         | 38.0                                         |
> | 25                                     | 34.0                                         | -                                            |
> | 50                                     | 30.0                                         | **48.0**                                     |
>
> We will add a discussion of this result in Section 4.3 of the camera-ready version.
>
> > Practitioners would highly benefit from guidelines for situations in which this approach is applicable vs where it is not
>
> As discussed in Section 4 and Appendix D.2, achieving strong performance with STL requires high action diversity (a large number of possible actions at each step). When action diversity is low, the effect of the value model on search performance is diminished, and it is necessary to roll out more tasks to obtain enough data to fine-tune the value model. For example, Section 4.2 (lines 202 - 204) notes that due to its low action diversity, HotpotQA shows smaller relative gains from STL compared to WebShop and also requires rollouts of 500 tasks (10 times more than WebShop) to get a large enough dataset for fine-tuning.
>
> Additionally, as you mentioned, STL requires later states to provide good value estimates that can be backed up and learned during training on lookahead results. Search performance will benefit most from STL on tasks where state transitions are *consistent throughout the environment*, i.e., the same or semantically similar actions yield similar outcomes. For instance, clicking the “Low to High” button on any search results page consistently orders items by price. This consistency enables the STL value model to accurately simulate a step of lookahead (Section 3.3), leading to better state-value estimation and improved downstream search performance. Tasks that have stochastic transitions or have inconsistent transitions when actions are semantically similar may not show large improvements with STL. Fortunately, most popular tasks where LLM agents have been deployed, such as web navigation, have deterministic and fairly consistent transitions.
>
> We agree that this could be useful information for practitioners and will add this discussion to Section 3 and the Limitations section in the camera-ready version of the paper.
>
>
> > The discussion of the presence/absence of human demonstrations in Figure 1 feels extraneous
>
> Thank you for the suggestion. We will remove the depiction of human demonstrations in Figure 1 and also rename the Reward and Demo Learning setting to Reward Learning in the camera-ready version.
>
> > Missing a discussion of train-time token usage when purely discussing test-time compute usage.
>
> Training data generation costs are quite modest, e.g., generating WebShop data with STL using a gpt-3.5-turbo policy and a llama-3.1-8b-instruct base value model costs $8.54. We note that this is a *fixed, one-time cost* which does not scale with the number of inference tasks. We will add a small discussion of train-time costs in Section 5.1 of the camera-ready version.
>
> > How did you obtain the Human Expert row in Table 1?
>
> These results were taken from the human expert evaluation performed in the original WebShop paper [2]. We will make this clear in the revised version.
>
> [1] Noah Shinn et al. 2023. “Reflexion: Language Agents with Verbal Reinforcement Learning.” Advances in Neural Information Processing Systems 36 (NeurIPS 2023), Main Conference Track
>
> [2] Shunyu Yao et al. 2023. “Tree of Thoughts: Deliberate Problem Solving with Large Language Models”. Advances in Neural Information Processing Systems 36 (NeurIPS 2023) Main Conference Track

---

> ### Comment · Reviewer_TNro · 2025-08-04
> **Thank you for your thorough rebuttal--it has addressed many of my questions**
>
> > We will update these results in the camera-ready version of the paper and state the changes made to the original repository to achieve these results in the Appendix. We note that this change does not affect the conclusions of the paper since Reflexion and STL belong to different reward settings.
>
> Thanks for tracing these to the bugs, your new results make sense
>
> > Therefore, since LATS and Reflexion involve a different level of supervision compared to STL, they are not true baselines, but rather help contextualize results of the Reward and Demo Free setting with the existing literature.
>
> Fully agreed, and with the bugs addressed I think the new pseudo-baselines better contextualize your results!
>
> > After some investigation, we believe that this early decline in unseen task performance is due to the limited number of tasks the value model sees during training during the first couple of iterations.
>
> Thank you for running this experiment--the results make sense.
>
> These experiments have clarified my concerns about both declining task performance and the quality of the baseline numbers. After reading this rebuttal, I've raised my score from a 3 to a 5.

---

> > ### Author Response · Authors · 2025-08-04
> >
> > Thank you for the thoughtful follow-up and for raising your score to a 5. We are glad that the additional experiments and the clarified baseline comparisons addressed your concerns and helped better contextualize our contributions. As we mentioned in the rebuttal, we will incorporate the updated discussion in the camera-ready version of the paper.

---

### Author Response · Authors · 2025-08-08
**Summary of Author-Reviewer Discussion**

We would like to thank the reviewers for their thoughtful feedback and engagement, which helped improve our paper. As the discussion period concludes, we would like to summarize the main points from the reviews, the rebuttals, and the ensuing discussion:

- **Discussion with TNro**:
  TNro noted a discrepancy in the Reflexion baseline. We reformatted the prompts to a conversational style and fixed a bug previously acknowledged by the original Reflexion paper's authors. These updates brought the baseline Reflexion performance in line with the ExpeL paper cited by TNro. They also asked about the performance drop in Figure 3, which we attributed to fewer tasks per iteration and resolved through additional experiments. TNro noted our rebuttal clarified their concerns and **raised their overall rating from 3 to 5**.

- **Discussion with rcW5**:
  rcW5 suggested an evaluation with newer baseline models. We added a greedy search baseline using a qwen3-8b value model, which showed similar trends to other greedy search baselines. We also clarified STL’s dependence on accurate natural language action descriptions when dealing with large action spaces, a discussion we will add to the camera-ready version of the paper. rcW5 confirmed all concerns were addressed, **kept their overall rating of 5**, and **increased their confidence from 4 to 5**.

- **Discussion with XxXy**:
 Following XxXy's recommendation, we added ReAct baselines using gpt-3.5-turbo and gpt-4o. As XxXy noted, these numbers *make the STL results "even stronger"*. They also suggested that providing more explicit examples illustrating why state transitions are easy to simulate while ground truth rewards are difficult to obtain would help clarify the motivation. We will revise the introduction accordingly in the camera-ready version. XxXy confirmed their concerns were fully addressed, **maintained their overall rating of 5**, and **raised their confidence from 3 to 4**.

- **Discussion with C12m**:
 C12m requested clarification on tree search and the STL rationale structure. We cited our original manuscript and included an example of the STL rationale structure in the rebuttal. C12m also suggested adding newer methods and models. As noted in the summary of discussion with rcW5, during the rebuttal, we evaluated greedy search using a qwen3-8b value model. While C12m has not yet responded, we would be happy to address any further questions or concerns.

---

### Decision · Program_Chairs · 2025-09-17

**Decision:**

Accept (spotlight)

**Comment:**

This paper proposes to improve LLM-based value model via Self-Taught Lookahead (STL). STL conducts tree search using state transition dynamics and backup value estate from rollout state to obtained better value estimates. The authors tested the proposed method on Webshop, HotPotQA, and Game-of-24 and find STL can improve a 8B open-weight value models to match the performance of gpt-4o on these tasks. Overall reviewers are happy with the significance, clarity and originality of the work. While there were some questions and confusions in the initial reviews, the authors conducted a very effective rebuttal, and addressed most of the concerns. Three reviewers voted to accept the paper as is. Reviewer C12m remained on the fence. The authors did compare to a couple of method published in 2024, so hopefully that alleviate the concern on comparison to more recent methods. The suggestion to include more discussion on the structure and role of rationale in the proposed method is valid, please address in the final draft.